# The multi-functional reovirus σ3 protein is a virulence factor that suppresses stress granule formation and is associated with myocardial injury

Yingying Guo[1], Meleana M. Hinchman[1], Mercedes Lewandrowski[1], Shaun T. Cross[1,2], Danica M. Sutherland[3], Olivia L. Welsh[3], Terence S. Dermody[3,4,5], John S. L. Parker[1,2]*

1 Baker Institute for Animal Health, College of Veterinary Medicine, Cornell University, Ithaca, New York, United States of America, 2 Cornell Institute of Host-Microbe Interactions and Disease, Cornell University, Ithaca, New York, United States of America, 3 Departments of Pediatrics, University of Pittsburgh School of Medicine, Pittsburgh, Pennsylvania, United States of America, 4 Departments of Microbiology and Molecular Genetics, University of Pittsburgh School of Medicine, Pittsburgh, Pennsylvania, United States of America, 5 Institute of Infection, Inflammation, and Immunity, UPMC Children's Hospital of Pittsburgh, Pennsylvania, United States of America

* jsp7@cornell.edu

**Data Availability Statement:** RNA seq data generated in this paper: The raw sequencing datasets can be found on NCBI Sequence Read

## Abstract

The mammalian orthoreovirus double-stranded (ds) RNA-binding protein σ3 is a multifunctional protein that promotes viral protein synthesis and facilitates viral entry and assembly. The dsRNA-binding capacity of σ3 correlates with its capacity to prevent dsRNA-mediated activation of protein kinase R (PKR). However, the effect of σ3 binding to dsRNA during viral infection is largely unknown. To identify functions of σ3 dsRNA-binding activity during reovirus infection, we engineered a panel of thirteen σ3 mutants and screened them for the capacity to bind dsRNA. Six mutants were defective in dsRNA binding, and mutations in these constructs cluster in a putative dsRNA-binding region on the surface of σ3. Two recombinant viruses expressing these σ3 dsRNA-binding mutants, K287T and R296T, display strikingly different phenotypes. In a cell-type dependent manner, K287T, but not R296T, replicates less efficiently than wild-type (WT) virus. In cells in which K287T virus demonstrates a replication deficit, PKR activation occurs and abundant stress granules (SGs) are formed at late times post-infection. In contrast, the R296T virus retains the capacity to suppress activation of PKR and does not mediate formation of SGs at late times post-infection. These findings indicate that σ3 inhibits PKR independently of its capacity to bind dsRNA. In infected mice, K287T produces lower viral titers in the spleen, liver, lungs, and heart relative to WT or R296T. Moreover, mice inoculated with WT or R296T viruses develop myocarditis, whereas those inoculated with K287T do not. Overall, our results indicate that σ3 functions to suppress PKR activation and subsequent SG formation during viral infection and that these functions correlate with virulence in mice.

Archive (SRA) under BioProject PRJNA699030 (https://www.ncbi.nlm.nih.gov/bioproject/) and analysis scripts can be found at: https://github.com/scross92/sigma3_mutants_RNAseq_analysis.

**Funding:** This research was supported by grants AI121216 to J.S.L.P and AI038296 to T.S.D. from the National Institute of Allergy and Infectious Diseases. STC was funded through the Cornell Institute of Host-Microbe Interactions and Disease (CIHMID) and NIH grant T32 AI145821. The funders had no role in study design, data collection and analysis, decision to publish, or preparation of the manuscript.

**Competing interests:** The authors have declared that competing interests exist.

## Author summary

The σ3 protein of mammalian orthoreoviruses is a double-stranded RNA binding protein that has classically been thought to function by scavenging dsRNA within infected cells and thus prevents activation of cellular sensors of dsRNA such as the kinase PKR. Here we used mutagenesis to identify the region of σ3 responsible for binding dsRNA. Characterization of mutant viruses expressing σ3 proteins incapable of binding dsRNA show that contrary to expectation, dsRNA binding is not required for σ3-mediated inhibition of PKR. We show that one mutant virus (R296T) despite being deficient in dsRNA-binding can inhibit PKR and replicates similar to WT virus. In contrast, another mutant virus (K287T) that bears a σ3 protein that cannot prevent dsRNA-mediated activation of PKR induces stress granules in infected cells and replicates less efficiently than WT virus. In vivo, the K287T mutant is attenuated in its replication and unlike WT virus and the R296T mutant virus does not cause heart disease (myocarditis).

## Introduction

Viral infection induces host cell responses that act to inhibit viral replication. A strong catalyst for these host responses is the synthesis of double-stranded RNA (dsRNA) during viral replication. Binding of dsRNA to innate immune sensors such as Toll-like receptor 3, retinoic acid-inducible gene I (RIG-I), and melanoma differentiation-association protein 5 (MDA5) initiates signaling that leads to secretion of type I and type III interferons (IFNs) [1]. IFNs then bind to their cognate cell-surface receptors to induce signaling and transcription of a large suite of IFN-stimulated antiviral genes [2].

Viral dsRNA produced during infection also can bind to other cellular innate immune sensors. Two well-characterized dsRNA-binding sensors are protein kinase R (PKR) and $2'-5'$-oligoadenylate synthase I (OAS). Following binding to dsRNA, these sensors activate signaling pathways that lead to inhibition of translation initiation and degradation of mRNA [3,4]. PKR is one of five cellular kinases, general amino acid control nonderepressible 2 (GCN2), heme-regulated eIF2α kinase (HRI), microtubule-affinity regulating kinase 2 (MARK2), PKR, and PKR-like endoplasmic reticulum kinase (PERK), that can phosphorylate the α subunit of eukaryotic initiation factor 2 (eIF2) [5–7]. Phosphorylation of eIF2α leads to sequestration of the GDP-GTP exchange factor eIF2B in an inactive GDP-bound complex with eIF2. This sequestration rapidly decreases the concentration of the eIF2•GTP•methionyl-initiator tRNA ternary complex (TC). As a consequence, translation initiation is inhibited, and the integrated stress response is activated [5].

Another consequence of PKR activation is the assembly of cytoplasmic ribonucleoprotein (RNP) stress granules (SGs) [8]. Cellular and viral mRNAs that stall on 43S ribosomes because of decreased availability of TCs are sequestered in SGs [9]. Cytosolic pathogen-recognition receptors such as PKR, MDA5, and RIG-I concentrate in SGs and, thus, SGs are hypothesized to act as signaling hubs for cellular antiviral responses [10]. Many viruses have evolved mechanisms to block or overcome the activation of PKR and prevent SG formation. For example, influenza A virus nonstructural protein 1 (NS1) binds to PKR and blocks its activation, thereby preventing PKR-mediated phosphorylation of eIF2α and SG formation [11]. Poliovirus strains block SG formation by 3C-mediated cleavage of the SG-nucleating protein, GTPase-activating protein-binding protein 1 (G3BP1) [12]. SGs are not the only type of RNP-containing granules that can form in cells in response to dsRNA. While SGs are the most well characterized, RNP granules that are RNase L-dependent and PKR-independent also form in cells treated with

poly I:C (a dsRNA mimic). These granules are termed RNase L-dependent bodies (RLBs) and are proposed to be induced during viral infection [13]. Whether RLBs are induced during reovirus infection is not known.

Mammalian ortheoreoviruses (reoviruses) are nonenveloped viruses that package a genome of 10 segments of dsRNA enclosed in a double-layered capsid. The outer-capsid layer serves to attach the virion to host cells and mediates viral penetration of cell membranes. Outer-capsid proteins are removed during viral entry to deposit a viral core particle in the cytoplasm. Transcription of the viral genome begins when the RNA-dependent RNA polymerase packaged in the core particle is activated after removal of the outer-capsid proteins. Viral mRNAs are extruded through turrets at the five-fold axes of symmetry. During passage through these turrets, the viral mRNAs are capped by guanyltransferase and methyltransferase enzymes [14].

Because reoviruses contain a dsRNA genome, they may have evolved mechanisms to prevent dsRNA-mediated activation of host innate defense mechanisms. One mechanism in which reoviruses likely avoid activating these sensors is by synthesizing their dsRNA genome segments in a viral core particle following encapsidation of single-stranded viral mRNAs representing each genome segment [15]. Reoviruses also encode a protein that can bind dsRNA, the viral outer-capsid protein, σ3 [16]. Overexpression of wild-type (WT) σ3, but not a dsRNA-binding-deficient mutant of σ3, rescues the inhibition of translation induced by PKR overexpression [17]. In addition, overexpression of σ3 rescues replication of E3L-deficient vaccinia virus and VAI-deficient adenoviruses, demonstrating that σ3 can complement other PKR-inhibiting viral proteins [18,19]. These findings led to the hypothesis that σ3 absorbs dsRNA in infected cells, thereby preventing PKR activation and global translational inhibition. However, this model does not explain why some strains of reoviruses replicate less efficiently in PKR-deficient mouse embryonic fibroblasts (MEFs) [20], where dsRNA-activation of PKR cannot occur, or why reoviruses replicate less efficiently in MEFs that only express a non-phosphorylatable form of eIF2α, findings that suggest reovirus replication benefits from activation of the integrated stress response [21]. Moreover, distribution of σ3 to both the cytoplasm and nucleus and the enrichment of σ3 in ribosomal fractions suggest that σ3 plays as yet unknown roles during viral infection [22,23].

Mechanisms that promote efficient reovirus translation are poorly understood. Reovirus mRNAs are noncanonical, being non-polyadenylated with relatively short 5′ and 3′ untranslated regions (UTRs). Polyadenylation at the 3′ end of host mRNAs promotes efficient translation and protects the host mRNAs from degradation by 3′-exonucleases. However, despite being non-polyadenylated, reovirus mRNAs are efficiently translated [24]. Viral mRNAs with shorter 5′ UTRs are generally translated less efficiently than those with longer 5′ UTRs. In addition, translational control elements have been identified in the 3′ UTR of reovirus mRNAs [25]. During infection, levels of viral mRNAs increase during infection and, if viral mRNA degradation occurs, it does not prevent abundant viral translation. Infection with some reovirus strains (e.g., type 2 Jones [T2J], type 3 clone 87, etc.) at high multiplicity induces host-cell translational shutoff [26]. Virus-induced host shutoff by such strains depends on PKR and RNase L to different extents. However, infection with other strains (e.g., type 3 Dearing [T3D]) does not substantially impair host-cell translation. Strain-specific differences in translational shut-off segregate with the σ3-encoding S4 genome segment [26]. It has been hypothesized that host translational shut-off efficiency correlates with the concentration of free σ3 not complexed with its partner outer-capsid protein, μ1, in the cytoplasm of infected cells [27]. How reovirus escapes host translational shutoff and maintains its own translation in the presence of activated PKR and phosphorylated eIF2α is not understood.

Reovirus infection induces formation of SGs that can be detected between 2–6 hours (h) post-infection (pi). However, by 12 h pi, SGs are repressed and cannot be induced by

treatment with sodium arsenite (SA), a potent inducer of SG formation that activates the HRI stress kinase [28,29]. The early induction of SG formation depends on eIF2α phosphorylation but occurs in cells individually lacking PKR, PERK, GCN2, and HRI [28], leading to the suggestion that at least two eIF2α kinases are activated. The early induction of SGs does not require viral transcription or protein synthesis, as UV-inactivated virions induce SG formation [28]. Formation of SGs early in infection is hypothesized to provide a competitive advantage for translation of viral mRNA. This idea also is supported by the finding that treatment of cells with SA to induce SG formation prior to infection enhances reovirus replication [30]. As infection proceeds, SGs are dissipated, even though eIF2α remains phosphorylated. However, if translation is chemically inhibited during infection, SG disassembly is inhibited, suggesting that a newly made protein is involved in SG disassembly [29]. At later stages of viral infection, SG proteins such as T-cell intracellular antigen-1-related protein (TIAR) and G3BP1 localize to the periphery of viral factories (VFs) [31]. An interaction between viral nonstructural proteins μNS and σNS with G3BP1 is required for this G3BP1 re-localization [31,32]. Co-expressing μNS and σNS dramatically changes the distribution of SG proteins and interferes with the formation of SA-induced SGs [31]. These data suggest that μNS and σNS are involved in disassembly of SA-induced SGs. However, the mechanism underlying SG disassembly during virus infection remains unclear, and other viral and host factors may be involved.

In this study, we engineered a panel of σ3 mutants and identified six that lacked detectable dsRNA-binding activity. We recovered recombinant viruses expressing two of these σ3 mutants, K287T or R296T, in the type 1 Lang (T1L) background. Surprisingly, only the T1L-K287T mutant displayed impaired replication in A549, Caco-2, and bone marrow-derived dendritic cells (BMDCs) relative to WT virus. Moreover, T1L-K287T-infected A549 cells contained more viral mRNA but less viral protein, and this virus was incapable of repressing SG formation at late times post-infection. Using newborn mice, we found that T1L-K287T produces lower viral titers at sites of secondary replication such as the liver, lungs, spleen, and heart relative to WT and T1L-R296T. Furthermore, unlike WT and T1L-R296T, the K287T mutant does not cause myocarditis. Our results indicate that independent of its capacity to bind dsRNA, σ3 regulates PKR phosphorylation and SG dynamics during viral infection and that this effect is required for efficient viral replication and pathogenesis.

## Results

### Identification and characterization of dsRNA-binding defective mutants of σ3

The reovirus outer-capsid protein σ3 is hypothesized to prevent PKR activation by binding and sequestering viral dsRNA during infection. To test this hypothesis, we first engineered mutants of σ3 guided by the structure of the σ3 dimer and previously predicted dsRNA-binding residues [33] (Fig 1A). We replaced basic residues on the surface of σ3, including residues R236, R239, K291, and K293 that were previously reported to be required for dsRNA binding [23,34,35]. Mutants were prepared individually as single point mutations or in combinations (Fig 1A). Mutant proteins were expressed in mammalian cells and tested for the capacity to bind a dsRNA mimic in a poly I:C pulldown assay (Fig 1B, 1C and 1D). Three mutants (R236T, R239T, and K291T) expressed poorly and two (R208T and K293T) had moderately decreased expression levels (Fig 1D). Of those mutants that were expressed at levels comparable to WT σ3, four mutant proteins (K287T, R202T, R296T, and R326T) were incapable of binding poly I:C (Fig 1C and 1D). We further tested the K287T and R296T mutants for the capacity to bind purified viral dsRNA, which is considerably longer than poly I:C and might be predicted to uncover any weak dsRNA binding. Although WT σ3 bound biotinylated viral

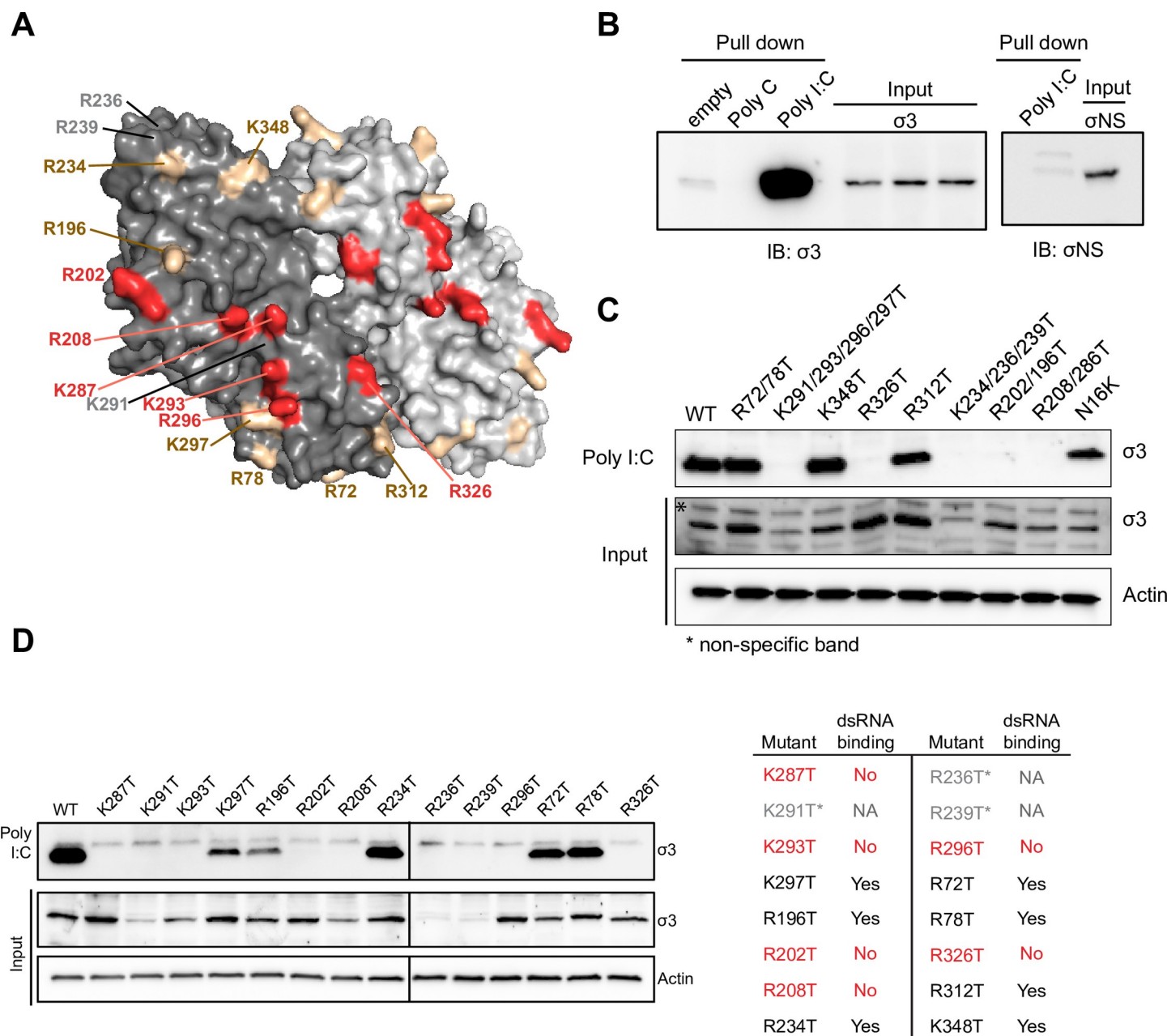

**Fig 1. Identification of double-stranded RNA-binding defective mutants of σ3.** (A) Space-filling surface view of the σ3 dimer structure (PDB ID: 1FN9) showing mutagenized positively charged residues (red and gold). Red indicates residues that disrupted the dsRNA-binding capacity of σ3. Gold indicates residues that retain their capacity to bind to dsRNA. Residues labelled in gray indicate those express poorly. The σ3 monomers are shown in different shades of gray. (B) Pull-down assay showing capacity of σ3, but not σNS to bind poly I:C. (C and D) Poly(I:C) pull-down assay showing capacity of σ3 mutants to bind dsRNA and relative expression levels when proteins were expressed exogenously. Findings are summarized in the table. An asterisk (*) and gray indicate mutants that express poorly. Red indicates residues that disrupted the dsRNA-binding capacity of σ3.

dsRNA efficiently, neither the K287T nor the R296T mutant displayed any detectable binding to biotinylated viral dsRNA (S1 Fig). Our findings are consistent with the prediction made by Olland et al. that a patch of basic residues distributed across the σ3 dimer interface is required for dsRNA binding [33] (Fig 1A, red colored residues).

When expressed alone, WT σ3 assembles as a dimer [33]. To confirm that the dsRNA-binding mutants are not impaired in the capacity to form dimers, we expressed N-terminally FLAG- and HA-tagged versions of each mutant and determined whether immunoprecipitation with an anti-FLAG antibody co-immunoprecipitated the HA-tagged version. All of the mutants were capable of dimerizing (S2A Fig). When σ3 is co-expressed with its partner outer-capsid protein, μ1, three copies of σ3 assemble with a trimer of μ1 to form a heterohexamer [36]. The surface of σ3 that interacts with μ1 is not near the residues required for dsRNA-binding. However, to ensure that mutations we introduced do not affect assembly of σ3-μ1 heterohexamers, we co-expressed μ1 and each of the WT or dsRNA-binding mutants of σ3 in HEK293 cells and assessed assembly of μ1 and σ3 by coimmunoprecipitation using a μ1-specific monoclonal antibody. All of the σ3 dsRNA-binding mutants tested coimmunoprecipitated with μ1, indicating that the assembly of μ1-σ3 heterohexamers is not impaired for the dsRNA-binding mutants (S2B Fig). These data demonstrate that other non-dsRNA-binding functions of σ3 remain intact.

## Recombinant σ3-mutant viruses are viable

To determine the function of dsRNA-binding during reovirus infection, we used reverse genetics [37] to engineer five recombinant viruses carrying the σ3 mutations that disrupt dsRNA-binding (R208T, K287T, K293T, R296T, and R326T) together with one mutant that retained the capacity to bind dsRNA (K297T) in the T1L and T3D backgrounds. All mutants were successfully recovered in L929 cells and sequence-confirmed to possess the predicted mutations. The mutants replicated with kinetics and produced peak titers comparable to WT virus in L929 cells during single- (S3A Fig) and multiple (S3B Fig) replication cycles, indicating that the lack of dsRNA-binding capacity by σ3 is not detrimental to replication in L929 cells.

## σ3 dsRNA-binding-defective mutant viruses do not activate PKR or eIF2α to any greater extent than WT virus in L929 cells

By sequestering dsRNA, σ3 is thought to prevent activation and autophosphorylation of PKR, thus preventing subsequent phosphorylation of eIF2α and inhibition of translation initiation [17–19,38]. Thus, we anticipated that viruses expressing mutant forms of σ3 incapable of binding dsRNA would activate PKR to higher levels. Although we observed no differences in replicative capacity of the mutants in L929 cells (S3 Fig), we wondered whether these viruses might have other mechanisms to counter the inhibition of translation initiation that occurs because of increased activation of PKR and downstream phosphorylation of eIF2α. We chose two dsRNA-binding mutants (K287T and R296T) of the five initially isolated to further characterize in detail based on their robust expression of σ3 in vitro. We, therefore assessed the levels of phosphorylated PKR and eIF2α in L929 cells that were mock-infected or infected with WT T1L or the dsRNA-binding mutants, K287T or R296T, in the T1L background. Surprisingly, we found that mock-infected L929 cells displayed constitutively phosphorylated PKR and high levels of phosphorylated eIF2α (S4 Fig). However, levels of phosphorylated PKR and eIF2α in WT-infected L929 cells were higher than those in mock-infected cells. Moreover, levels of phosphorylated PKR and phosphorylated eIF2α in L929 cells infected with K287T and R296T viruses were not substantially higher than those in the WT-infected cells (S4 Fig). Based on these findings, we conclude that L929 cells have constitutively phosphorylated PKR and that WT and mutant reoviruses replicate to high titers in these cells despite the presence of phosphorylated PKR and eIF2α. Activation of the integrated stress response, which occurs following eIF2α phosphorylation, appears to benefit reovirus replication [21]. In addition, pretreatment of L929 cells with SA, which promotes SG formation, enhances reovirus

replication [30]. The L929 cells used to propagate reoviruses are routinely grown in spinner culture and are highly susceptible to reovirus infection. Based on these findings, we conclude that our laboratory L929 cells, although susceptible to the σ3 dsRNA-binding mutants, would not be useful to discriminate potential effects of the mutant viruses on host responses to infection.

## The T1L-K287T mutant produces lower titers than WT virus in epithelial cells and bone marrow dendritic cells (BMDCs)

To determine whether the dsRNA-binding mutants display replication deficits in other cell types, we assessed their capacity to replicate in epithelial cells (A549 and Caco-2) and murine bone marrow-derived dendritic cells (BMDCs) (Fig 2A, 2B and 2C). A549 cells were infected with 10 PFU per cell of WT T1L or mutant viruses, and viral yields at 24, 48, and 72 h pi were determined. T1L-K287T titer was approximately 10-fold less than that of WT T1L at 24, 48, and 72 h pi. T1L-R296T titer did not significantly differ from WT T1L titer at any timepoint tested. Similarly, we found that T1L-K287T replicated less efficiently than WT T1L in Caco-2 cells (Fig 2B). In BMDCs, while all viruses replicated poorly, the T1L-K287T and T1L-R296T mutants replicated less efficiently than WT T1L (Fig 2C). Collectively, these data demonstrate cell type-specific replication deficits for K287T and R296T viruses, indicating different cellular responses to infection.

## Proteolytic processing of WT, T1L-K287T, and T1L-R296T viruses are comparable

During reovirus entry, the σ3 outer-capsid protein is proteolytically cleaved in endosomes and removed from the viral particle [39]. Removal of σ3 is required to activate the μ1 protein for cellular membrane penetration (reviewed in [40]). While the engineered σ3 mutations are not near previously identified cleavage sites in σ3, we thought it possible that proteolytic processing of σ3-K287T could be delayed. We used limited trypsin proteolysis to replicate endosomal disassembly steps and compared the kinetics of σ3 processing of purified virions (S5 Fig).

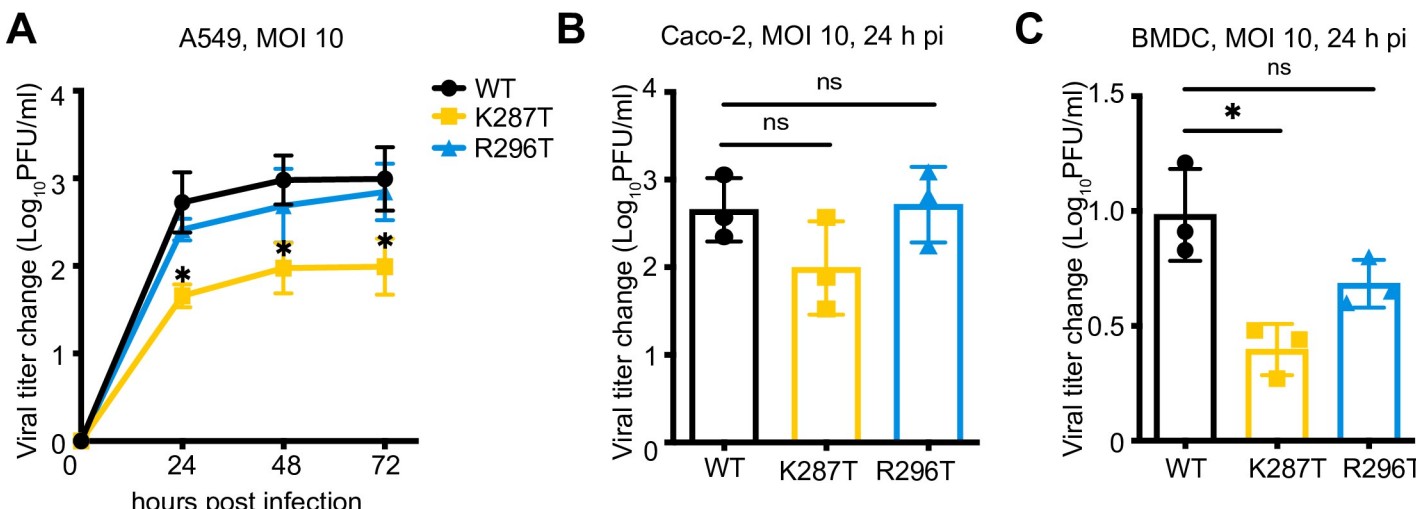

**Fig 2. T1L, T1L-K287T, or T1L-R296T viral replication in A549, Caco-2, and bone marrow-derived dendritic cells (BMDC).** A549 (A) or Caco-2 (B) or BMDC (C) were infected with the indicated viruses at 10 PFU per cell. Plaque assays were used to assess viral yield. Data are reported as mean ± s.d., n = 3 independent experiments. For (A), P values were calculated to analyze the differences comparing with T1L WT using a two-way analysis of variance (ANOVA) followed by Tukey's test (* P < 0.05). For (B) and (C), P values were calculated using multiple non-paired t test. * P < 0.05, ns = not significant.

Kinetics of σ3 processing for the WT, K287T, and R296T viruses were comparable, making it unlikely that differences in viral entry kinetics explain the differences in replication we noted in A549 cells and BMDCs. The particle-to-PFU ratios for purified WT, K287T, and R296T virions were 184, 430, and 158, respectively and fall within the reported range for reovirus [41,42].

## The K287T mutant virus synthesizes more viral mRNA, but less viral protein and less negative-sense viral RNA than WT virus

To further investigate mechanisms underlying the replication deficit of the K287T mutant in A549 lung epithelial cells, we infected cells with 100 PFU per cell of WT T1L or the σ3 mutants and assayed levels of viral mRNA, viral protein, and negative-sense RNA. Despite producing lower yields than WT virus, K287T produced 2.5 to 4-fold higher levels of viral mRNA than WT (Fig 3A). However, the higher levels of viral mRNA in K287T-infected cells were accompanied by lower levels of viral proteins than those in WT-infected cells (Fig 3B and 3C). Negative-sense viral RNA accumulation is an indirect measure of dsRNA synthesis, as reoviruses package mRNA into viral core particles prior to replication. Consistent with reduced viral titers, cells infected with the K287T mutant contained ~ 4-fold less negative-sense s4 RNA than WT-infected cells (Fig 3D). The R296T mutant also synthesized 1.5 to 3-fold more viral mRNA, had decreased protein expression, and synthesized ~ 25% less negative-sense s4 RNA relative to WT virus, but these differences were not statistically significant (Fig 3A, 3B, 3C and 3D).

The findings of diminished protein expression suggested that translation in K287T- and R296T-infected cells is inhibited, and the findings that viral mRNA levels are increased suggested that viral mRNA is sequestered and protected from normal turnover and translation. Therefore, we assessed phosphorylation of PKR and eIF2α in A549 cells that were either mock-infected or infected with T1L, K287T, and R296T (Fig 3E and 3F). While cells infected with the K287T mutant displayed increased phosphorylation of PKR and eIF2α, levels of phosphorylated PKR and eIF2α in WT- and R296T-infected cells did not differ appreciably from those in mock-infected cells (Fig 3E and 3F). Collectively, these data suggest that the capacity of σ3 to bind dsRNA is not required to suppress the activity of PKR, as evidenced by near WT replication, protein synthesis, and viral RNA synthesis of the R296T mutant. However, these data additionally suggest that some mutations that ablate the capacity of σ3 to bind dsRNA appear to overlap with a region of the protein responsible for blocking the activity of PKR.

The increased phosphorylation of eIF2α and decreased viral protein synthesis in cells infected with the K287T mutant suggested that cellular translation is globally repressed. To test this hypothesis, we used SUnSET puromycin labeling [43,44] to quantify global protein synthesis in A549 cells infected with WT, K287T, or R296T relative to mock-infected cells. In cells treated with poly I:C, we noted strong inhibition of incorporation of puromycin into elongating polypeptides compared with untreated cells (S6A Fig). We found that the overall puromycin incorporation in mock-, WT-, and R296T-infected cells was similar. In contrast, cells infected with K287T displayed significant suppression of global protein synthesis, consistent with inhibition of translational initiation induced by phosphorylation of eIF2α (S6B and S6C Fig).

## The T1L-K287T mutant virus induces abundant stress granule formation

Based on results presented thus far, we suspected that K287T (and perhaps to a lesser extent R296T) mRNA could not fully engage the translational machinery due to viral mRNA sequestration in stress granules (SGs). SGs are liquid-liquid phase-separated structures that form in

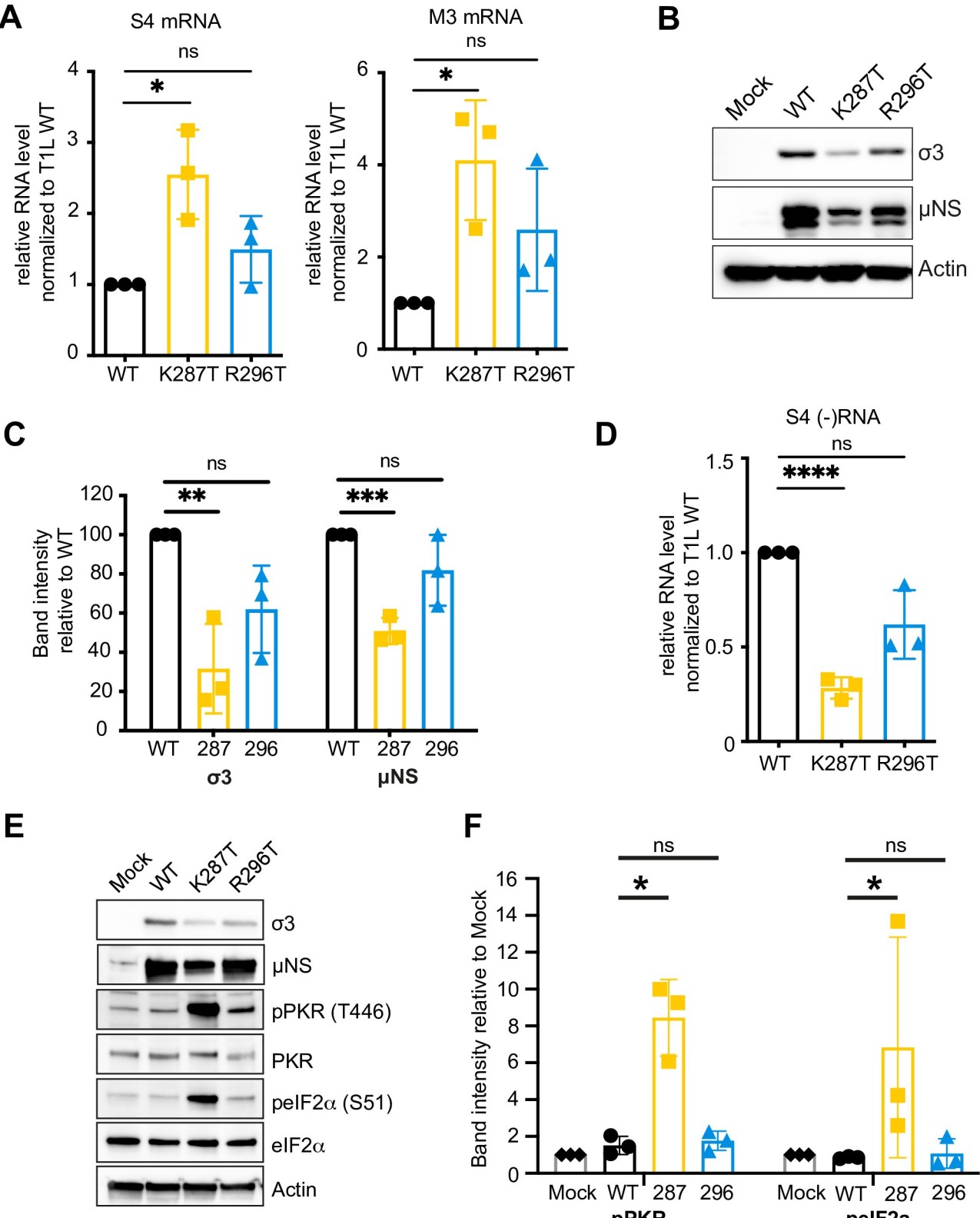

**Fig 3. A549 cells infected with T1L-K287T contain higher levels of viral mRNA than WT-infected cells despite lower levels of viral replication and lower levels of viral protein.** A549 cells were infected with T1L, either WT or K287T or R296T, at 100 PFU per cell. At 18 h pi, the levels of viral (A) s4 and m3 mRNA were assayed by RT-qPCR. Relative RNA levels were first normalized to GAPDH levels and then WT RNA levels were set as 1.

(B) Representative immunoblot showing the relative viral protein expression for each virus; quantification in (C). (D) Relative levels of s4 negative-sense RNA assayed by RT-qPCR. (E) Representative immunoblot showing levels of phosphorylated PKR and phosphorylated eIF2α in mock-infected and WT-, K287T- and R296T-infected A549 cells at 18 h pi; quantification in (F). Data are presented as mean ± s.d. of three independent experiments. Non-paired $t$ tests were used to evaluate statistical significance. * $P < 0.05$, ** $P < 0.01$, *** $P < 0.001$, **** $P < 0.0001$, ns = not significant.

the cytoplasm of cells under conditions of stress, where mRNAs bound to stalled 48S translation initiation complexes are sequestered along with SG markers such as TIAR and G3BP1 (reviewed in [8]). SGs form early after reovirus infection with the T2J and T3D strains (as early as 2 h pi), peak at 6 h pi, and disappear later in infection (24 h pi), at which time reovirus-infected cells are resistant to exogenous induction of SGs [28,29]. Qin et al [28] found that infection with virions of the T1L strain does not induce SGs in the cell lines tested. The authors hypothesized that this difference is attributable to differences in the kinetics of cell entry and found that following infection with T1L infectious subvirion particles, SGs formed at levels similar to those induced by the T2J and T3D strains [28]. To test for differences in SG formation, we infected cells with WT or mutant virions and assayed reovirus infection and TIAR-positive puncta at 6 and 18 h pi using immunofluorescence microscopy (Fig 4A). At 6 h pi, ~ 2–3% of all virus-infected cells contained TIAR-positive granules (Fig 4B). However, by 18 h pi, significantly more K287T-infected cells contained TIAR-positive granules (mean, 46.4%) than WT- (mean, 2.6%) or R296T-infected cells (mean, 5.0%) (Fig 4B). To further confirm that the TIAR-positive granules observed in K287T-infected cells were SGs, we co-immunostained for another SG marker protein, G3BP1. We found that the TIAR and G3BP1 markers often co-localized in cytoplasmic granules of K287T-infected cells (Fig 4C). Interestingly, G3BP1 and TIAR co-localized to some degree with μNS, a marker for reovirus replication sites or factories. Based on these findings, we hypothesized that the T1L-K287T virus is defective in suppressing the formation of SGs.

## Viral mRNA partially localizes to SGs in K287T- and R296T-infected cells

Excess viral mRNA in K287T-infected cells and abundant accumulation of SGs at late times post-infection prompted us to evaluate the intracellular distribution of viral mRNA. We used smFISH to visualize s4 viral mRNA in mock-, WT-, K287T-, and R296T-infected cells and co-immunostained cells with markers for SGs (TIAR) and VFs (λ2). In mock-infected cells, as expected, control *GAPDH* mRNA was distributed throughout the cytoplasm (S7 Fig). In WT-infected cells, *GAPDH* mRNA distribution appeared diffuse and similar to that in mock-infected cells (S7 Fig), whereas s4 mRNA was concentrated in VFs, indicating that viral mRNAs, but not cellular mRNAs, localize to VFs (Figs 5 and S7). As in previous experiments, few TIAR-positive SGs were detected in WT-infected cells. In K287T-infected cells, s4 mRNA co-localized with the marker for VFs. In addition, K287T s4 mRNA co-localized with TIAR-positive SGs (Fig 5). Occasionally (~ 7% of cells), we observed triple co-localization of TIAR, s4 mRNA, and VFs (λ2) in K287T-infected cells (Fig 5). Notably, VFs in K287T-infected cells were usually small. In R296T-infected cells, s4 mRNA co-localized with VFs. However, s4 mRNA also co-localized with TIAR-positive SGs when they were present. Collectively, these data demonstrate that SGs in K287T- and R296T-infected cells contain viral mRNA and suggest that eIF2α phosphorylation promotes translational inhibition and the sequestration of viral mRNA within SGs.

Based on the localization of SG marker TIAR and the viral s4 mRNA and λ2 protein, we classified infected cells into four phenotypes based on the localization of s4 mRNA relative to λ2-labeled VFs and TIAR-positive SGs. The four phenotypes we identified were cells in which (a) s4 mRNA co-localized within VFs as detected by λ2 staining; (b) s4 mRNA co-localized

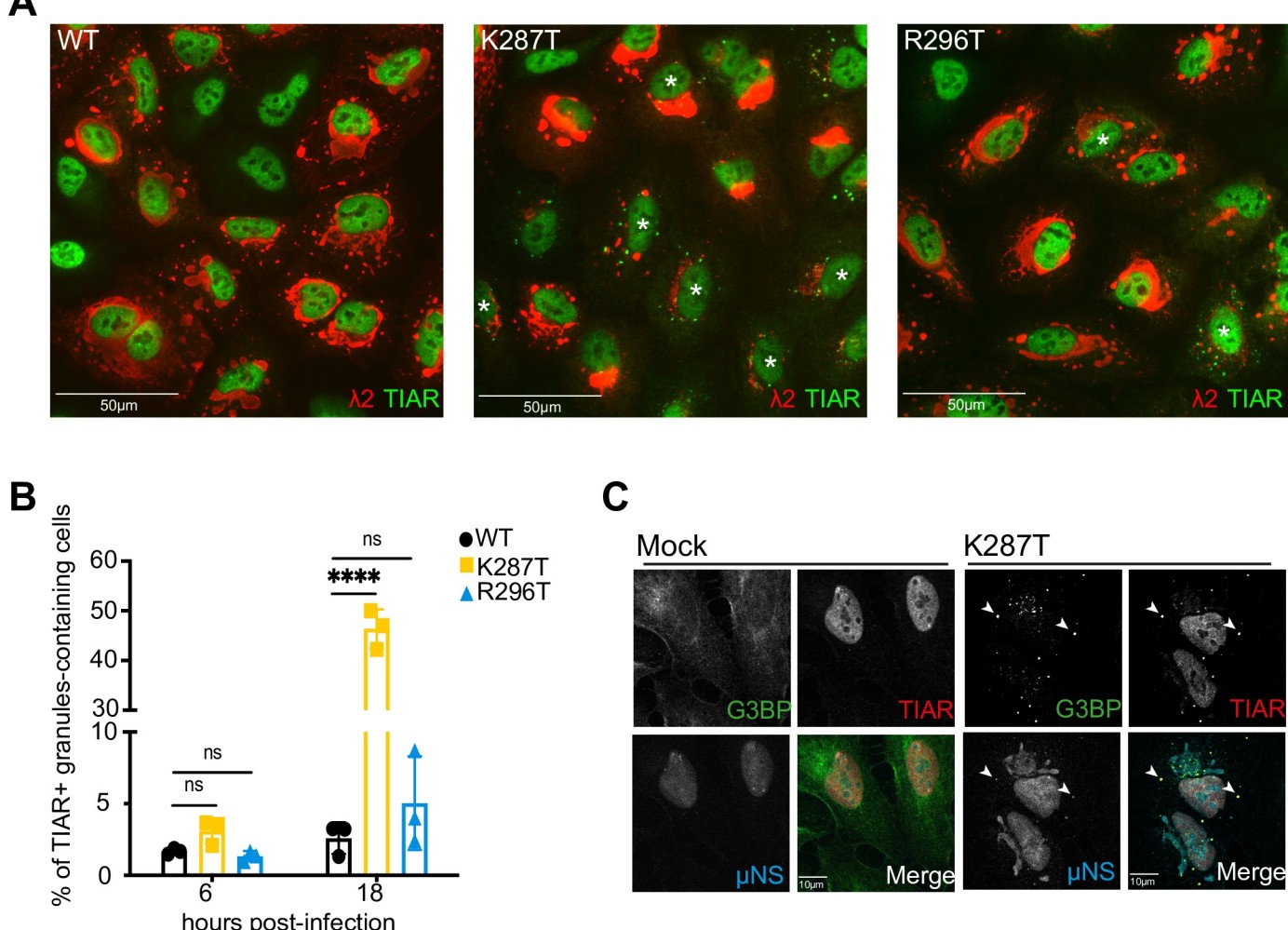

**Fig 4. Significantly more A549 cells infected with T1L-K287T contain TIAR-positive RNP granules at 18 h pi than cells infected with T1L-WT or -R296T.**
(A) Cells were infected at 100 PFU per cell with T1L-WT, -K287T, or -R296T. At 18 h pi, cells were fixed and immunostained with mouse anti-λ2 monoclonal antibody (7F4) and anti-TIAR antibody. White asterisks show infected cells containing TIAR-positive granules. Note: the brightness and contrast of the red channel of the immunofluorescence image of the K287T-infected cells has been enhanced to allow low levels of protein expression within some of the K287T-infected cells to be seen. (B) Percentage of infected cells containing TIAR-positive granules at the indicated time points. A minimum of 200 cells were counted for each experiment. Data shown represent the mean ± s.d. of three independent experiments. Multiple comparison, non-paired *t* tests were used to analyze differences (ns = not significant, **** $P < 0.0001$). (C) Mock- and T1L-K287T-infected A549 cells at 18 h pi showing co-localization of TIAR, G3BP and μNS in some infected cells. White arrows indicate co-localization of these three proteins. Note the μNS antibody shows non-specific background staining of cellular nuclei.

with TIAR in SGs; (c) s4 mRNA, TIAR, and λ2 co-localized together in VFs/SGs; and (d) both phenotypes (a) and (b) were present in the same cell. We then quantified the mean fluorescence levels of viral s4 mRNA and λ2 protein for each cell (S8 Fig). A total of 25, 26, and 66 WT-, R296T-, and K287T-infected cells were analyzed. Overall, we found that the mean λ2 protein level per cell for K287T-infected cells was significantly lower than that for WT- and R296T-infected cells (S8 Fig). When the mean fluorescence levels were examined on the basis of the four cell phenotypes, we found that λ2 protein mean fluorescence levels were significantly lower in K287T-infected cells that had SGs than in cells in which SGs were absent (S8 Fig). When we analyzed the mean per cell levels of s4 mRNA fluorescence, we found no difference in the mean levels of WT-, R296T-, and K287T-infected cells. However, K287T-infected cells that contained SGs had significantly more s4 mRNA per cell than K287T-infected cells

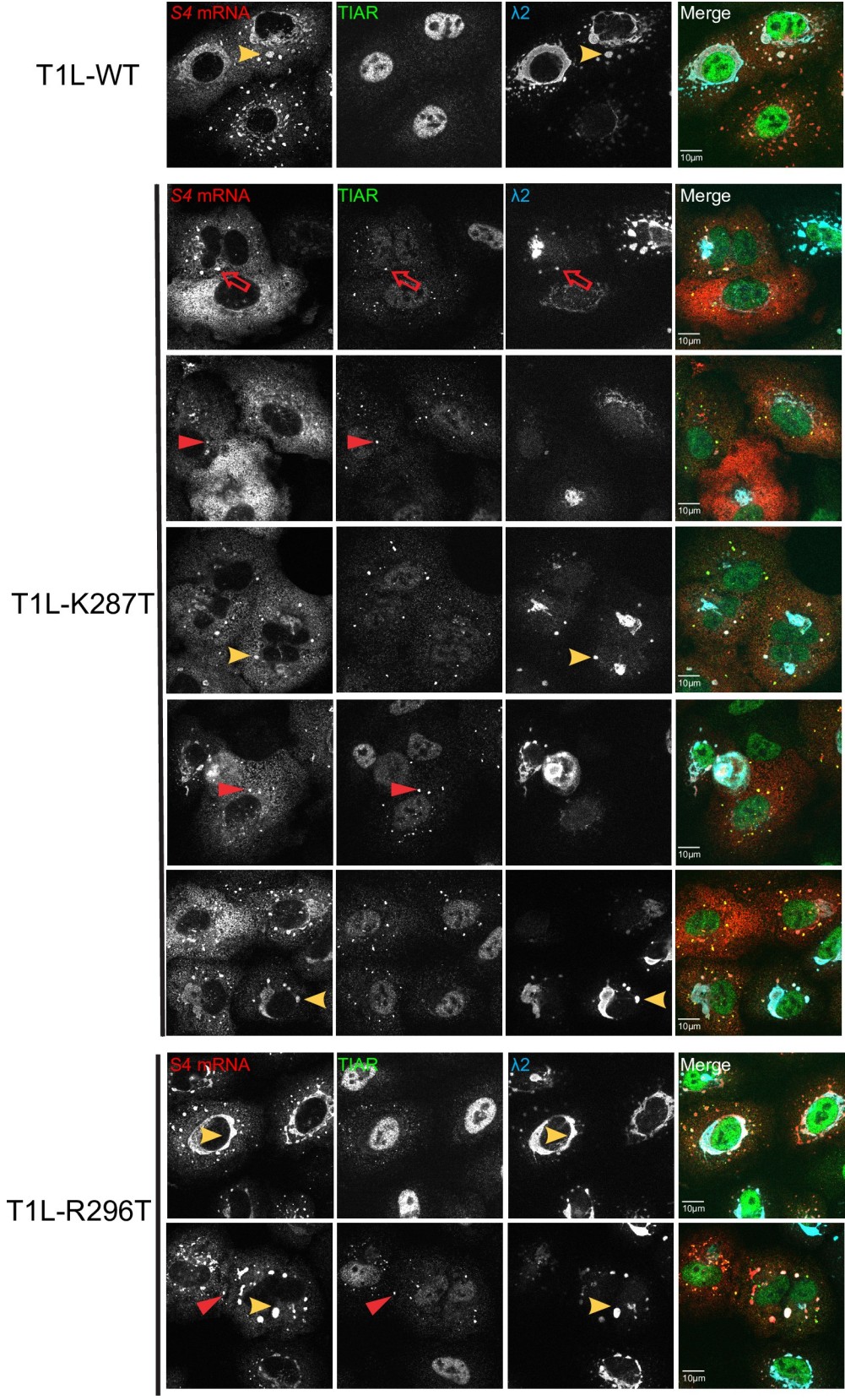

**Fig 5. Viral RNA is sequestered in TIAR-positive granules in cells infected with T1L-K287T.** Cells were infected at 100 PFU per cell with T1L-WT, -K287T, or -R296T. At 18 h pi, cells were fixed for immunofluorescence staining with antibodies against λ2 (7F4) and TIAR, followed by secondary antibodies staining. Subsequently, CAL Fluor Red 610 Dye-conjugated s4 mRNA probes were used to detect localization of s4 mRNA. Red arrowheads indicate colocalization of s4 mRNA and TIAR. Yellow arrowheads indicate colocalization of s4 mRNA and viral factories (λ2). Red unfilled arrows indicate triple co-localization of TIAR, s4 mRNA and viral factories (λ2). Images were collected using Olympus FLUOVIEW FV3000. Data are quantified in S8 Fig.

that lacked SGs. Moreover, all infected cells with SGs had significantly higher average viral s4 mRNA per cell and lower mean viral protein (λ2) per cell than infected cells without SGs, independent of the virus used (S8 Fig). Based on these results, we conclude that K287T-infected cells that contained SGs had lower levels of viral λ2 protein and higher levels of s4 mRNA than WT- and R296T-infected cells and K287T-infected cells that did not contain SGs.

## Ribonucleoprotein (RNP) granule formation by the σ3-mutant viruses depends on PKR and RNase L to different extents

RNP granules called RNase L bodies (RLBs) due to their dependence on RNase L form in cells transfected with poly I:C, a dsRNA mimic [13,45]. RLBs are distinct from SGs in that they (i) are smaller and more punctate, (ii) are dependent on RNase L but form independently of PKR activation and eIF2α phosphorylation, and (iii) differ in their protein composition [13,45]. Treatment of A549 cells with poly I:C induces formation of RLBs, rather than SGs, due to the repression of SGs by activated RNase L [13]. The granules that form in reovirus-infected cells at early times post-infection (2–4 h pi) require viral core particles to access the cytoplasm of newly infected cells as well as phosphorylation of eIF2α, but their formation does not require viral transcription or translation of viral mRNAs [28], suggesting that these granules are SGs. RNP granules in reovirus-infected cells form in the absence of PKR, HRI, PERK, or GCN, suggesting that at least two of the stress kinases that phosphorylate eIF2α are involved in the formation of SG after entry of the reovirus core particle into the cytoplasm [28]. Alternatively, the granules observed in PKR knock-out (KO) MEFs are RLBs. After viral replication begins, dsRNA accumulates in infected cells and could induce formation of RLBs, or dsRNA could be released from core particles damaged during viral entry [46]. The marker proteins used thus far, TIAR and G3BP1, localize to both RLBs and SGs (S9 Fig). To determine whether the TIAR/G3BP1-positive granules detected in K287T- and R296T-infected A549 cells at 18 h pi are SGs or RLBs, we first assessed the requirement for PKR and RNase L in granule formation using A549 cells lacking PKR, RNase L, or both. As expected, TIAR-positive RNP granules were rare or absent in cells infected with WT virus at 18 h pi (Fig 6A). In contrast, infection with the K287T virus produced TIAR-positive granules in the cytoplasm of 60.0 ± 7.8% and 58.2 ± 6.6% of WT and RNase L KO cells, respectively (Fig 6A), indicating that the granules formed independently of RNase L. However, 8.9 ± 5.4% of K287T-infected PKR KO cells contained TIAR-positive granules, indicating that formation of most granules was PKR-dependent, while some granules formed independently of PKR and were likely RLBs. TIAR-positive granules were completely absent in infected double-knockout (DKO) cells. Notably, TIAR-positive granules were present in 14.6 ± 1.2%, 10.2 ± 1.6%, and 10.0 ± 2.6% of WT, PKR KO, and RNase L KO cells infected with the R296T mutant, respectively. Again, no RNP granules were detected in DKO cells. These findings suggest that the R296T mutant differs from the K287T mutant in that it suppresses PKR-dependent granule formation more efficiently. Taken together, these findings suggest that σ3 functions to suppress both SGs and RLBs during infection, but that PKR-dependent SGs are the predominant form of RNP granule that is induced and suppressed by σ3 in reovirus-infected cells.

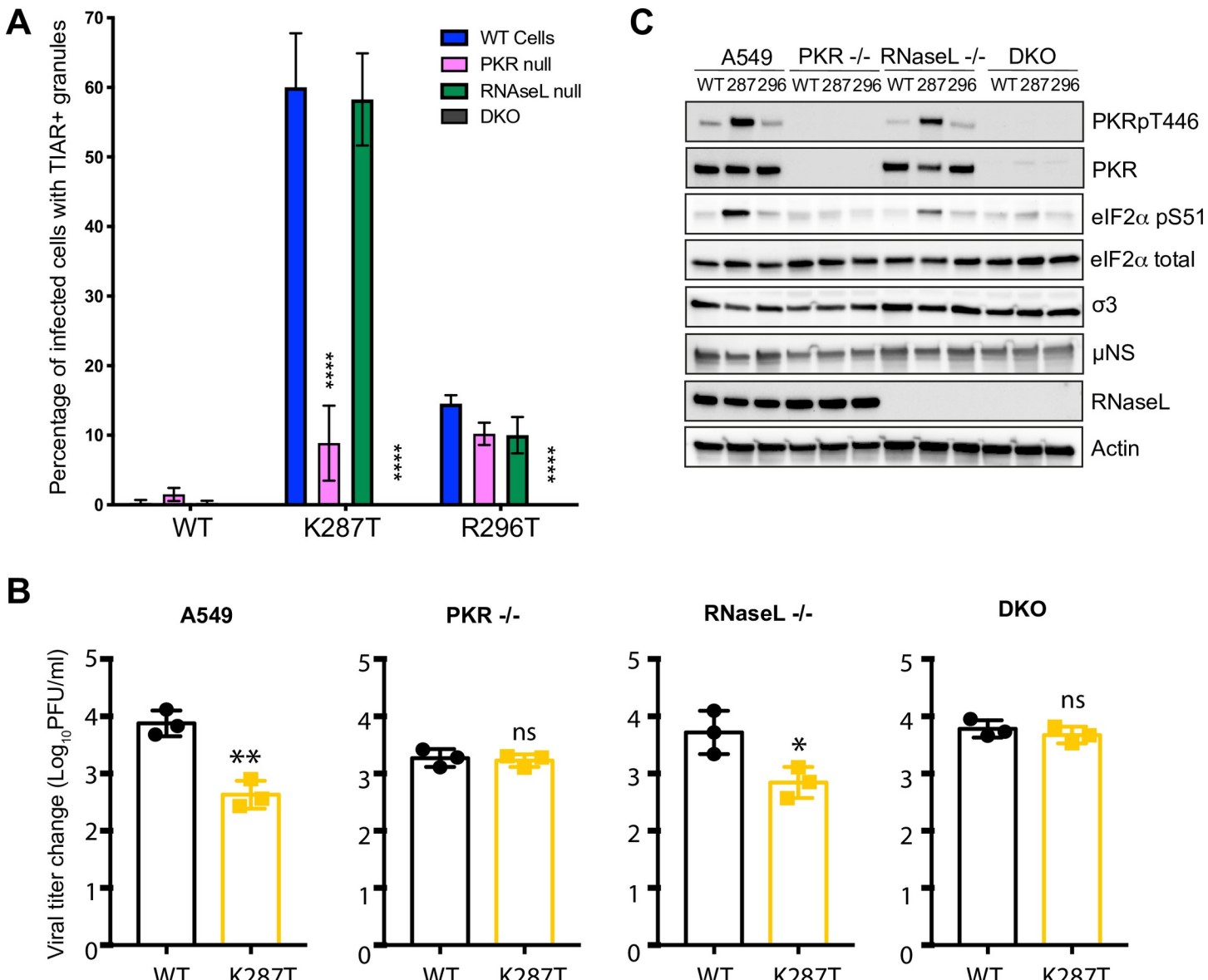

**Fig 6. T1L-K287T does not suppress activation of PKR and PKR-dependent stress granule (SG) formation.** (A) Formation of TIAR-positive granules in A549 parental, PKR KO, RNase L KO and DKO A549 cells at 18 h pi. The cell lines shown were infected with WT or T1L-K287T or T1L-R296T mutant viruses (100 PFU per cell). At 18 h pi, cells were fixed for immunofluorescence staining with antibodies against TIAR and viral protein λ2 and the number of infected cells containing > 3 foci of TIAR-positive granules were counted. Data are expressed as the percent of infected cells containing TIAR-positive granules (n = 3, > 200 infected cells counted in 10 randomly selected fields). Data shown represent the mean ± s.d. of three independent experiments. *P* values were calculated to analyze the differences comparing with A549 parental cells using a two-way analysis of variance (ANOVA) followed by Tukey's test (**** *P* < 0.0001). (B) Viral replication in A549 parental, PKR KO, RNase L KO and DKO cells. Cells were infected with the indicated viruses (10 PFU per cell). Viral growth is shown as change in viral titer from T = 0 to 24 h pi. Data shown represent the mean ± s.d. of three independent experiments. Multiple comparison, non-paired *t* tests were used to analyze differences (ns = not significant, * *P* < 0.05, ** *P* < 0.01). (C) PKR and eIF2α are hyperphosphorylated in T1L-K287T infected A549 cells. Cells were infected as described in (A). A representative immunoblot is shown.

To further delineate SG and RLB induction by reovirus, we used immunofluorescence detection of SG and RLB markers to characterize the granules present in K287T- and R296T-infected cells. No protein markers have been confirmed to uniquely mark RLBs. However, several proteins localize only to SGs (TIA1, FMRP and PUM1) and not RLBs [13]. We used TIA1 to uniquely label SGs and G3BP1 to label both SGs and RLBs. We confirmed that the presence

of TIA1 distinguishes SGs from RLBs by treating WT, PKR KO, RNase L KO, and DKO A549 cells with poly I:C or SA. As previously shown, poly I:C treatment of WT or PKR KO cells led to the formation of G3BP1-positive/TIA1-negative RLBs and the formation of G3BP1-positive/TIA-1-positive SGs in RNase L KO cells [13] (S10 Fig). SA treatment, which activates the stress kinase HRI, induced SGs in all cell lines, indicating that this pathway of SG formation does not require PKR or RNase L (S10 Fig). Having confirmed the specificity of these markers, we analyzed patterns of TIA1 and G3BP staining in K287T- and R296T-infected cells (S11 Fig). We found that K287T-infected WT and RNase L KO cells contained TIA1/G3BP1 double-positive granules, indicative of SGs. However, in infected PKR KO cells, the granules were TIA1-negative/G3BP1-positive, indicating that they were RLBs. As before, no granules were detected in DKO cells. In R296T-infected WT cells, the granules were TIA1-negative/G3BP1 positive, indicative of RLBs. However, a few cells had a mixture of TIA-1/G3BP1-double positive granules and TIA1-negative/G3BP1-positive granules, suggesting that the R296T mutant mostly induces RLBs but does not completely suppress SGs. Consistent with this hypothesis, the granules that formed in R296T-infected RNase L KO cells were TIA1/G3BP1 double-positive granules, indicative of SGs. Collectively, these data demonstrate that the K287T virus predominantly induces PKR-dependent SGs, while the R296T virus mostly induces RLBs.

## Replication deficit of K287T is rescued in PKR-deficient cells

Because TIAR-positive granule formation at 18 h pi appeared dependent on both PKR and RNase L, we investigated whether loss of PKR, RNase L, or both would rescue the viral replication deficit of K287T. We infected WT, PKR KO, RNase L KO, or DKO A549 cells with 10 PFU per cell of WT or the K287T σ3-mutant virus and quantified viral yields at 24 h pi. In WT cells, we confirmed that the K287T mutant produced significantly less virus at 24 h pi than WT virus (Fig 6B). However, in PKR KO cells, yields of WT and K287T did not differ significantly. In addition, the K287T mutant produced to ~ 3-fold higher titer in PKR KO cells than in WT A549 cells. Notably, yields of WT virus in PKR KO cells were ~ 5-fold less than yields in WT A549 cells, consistent with previous findings that lack of PKR diminishes yields of some reovirus strains [20]. In RNase L KO cells, the K287T mutant produced ~ 10-fold less virus than WT and showed a similar replication pattern to that in WT cells (Fig 6B). The yield of WT virus in RNase L KO cells was similar to its yield in WT A549 cells. In DKO cells, WT and K287T produced comparable yields (Fig 6B). Interestingly, despite the lack of PKR in DKO cells, WT virus produced yields similar to those in WT cells, suggesting that any growth defect due to lack of PKR is overcome when RNase L is also absent. Collectively, these data indicate that PKR impairs replication of the K287T mutant.

## PKR and eIF2α are hyperphosphorylated in A549 cells infected with K287T

Since K287T infection of PKR KO cells produced 6-fold fewer RNP granule-containing cells than infection of WT cells, we hypothesized that PKR and eIF2α phosphorylation are required to form SGs in K287T-infected cells. We therefore examined levels of phosphorylated PKR and eIF2α in WT-, K287T-, and R296T-infected WT, PKR KO, RNase L KO, and DKO A549 cells. We observed abundant phosphorylation of both PKR and eIF2α in K287T-infected WT and RNase L KO cells (Fig 6C). In contrast, T1L and R296T produced comparable and minimal levels of phosphorylated PKR and eIF2α, independent of the cell type used (Fig 6C). These data indicate that PKR and eIF2α phosphorylation correlates with SG formation in K287T-infected cells.

## Ectopic expression of σ3 can suppress poly I:C-induced SGs in RNase L-deficient A549 cells but cannot suppress RNase L bodies in WT A549 cells

Our findings thus far suggested that σ3 functions to inhibit PKR and, as a consequence, prevents phosphorylation of eIF2α and formation of SGs in infected cells. However, others have shown that the viral μNS and σNS proteins act to suppress SG formation during infection, raising the possibility that a combination of σ3, μNS, and σNS act to prevent and suppress SG formation during infection. To test whether σ3 alone can prevent PKR phosphorylation and SG formation, we prepared stable doxycycline-inducible WT and RNase L KO A549 cells expressing WT, K287T, and R296T σ3. We used poly I:C treatment to induce PKR phosphorylation and found that expression of WT or R296T mutant σ3, but not K287T σ3 prevented poly I:C-induced PKR phosphorylation in WT A549 cells (S12A Fig). As previously reported [13], and shown in S9 Fig, poly I:C induces the formation of RLBs in WT A549 cells and SGs in RNase L KO cells. Therefore, we assessed the capacity of WT and mutant σ3 to suppress RLBs in WT cells (S12B Fig) and SGs in RNase L KO cells (S12C Fig). We found that the poly I:C-induced RLB formation in WT A549 cells was not suppressed by expression of WT, K287T, or R296T σ3 (S12B Fig). In contrast, SG formation in RNase L KO A549 cells was significantly suppressed by expression of WT and R296T σ3 but not by expression of K287T σ3 (S12C Fig). Based on these findings, we conclude that ectopic expression of σ3 is sufficient to prevent PKR phosphorylation, as previously reported [17,38]. In addition, σ3 expression can prevent SG formation but has no apparent effect on RLB formation.

## The K287T mutant is less virulent than WT virus in mice

Because the dsRNA-binding mutants displayed cell-type specific replication impairment and differences in stress-induced responses, we evaluated whether these mutants differ from WT virus in the capacity to cause disease in newborn mice. Neonatal C57BL/6 mice were inoculated perorally with WT, K287T, or R296T virus strains, and viral yields and gross pathology in various tissues were assessed. While all viruses produced similar titers in the intestine at 1 d pi, significant differences in K287T viral loads were detected at later times and in sites of secondary replication such as the spleen, liver, lungs, and in particular, the heart (Figs 7 and 8). While K287T titers in the heart were similar to WT and R296T viruses at 1, 2, and 4 d pi, a significant decrease in titers of this virus was observed at 8 d pi (Fig 7). Interestingly, there were no significant differences in viral titers observed in the brain, suggesting that viral dissemination is not altered and that K287T displays tissue-specific impairment at certain sites. Furthermore, in contrast to WT and R296T, 287T did not induce detectable myocarditis, which is inflammation of heart tissue associated with aberrant deposition of calcium in the myocardium [47] (Fig 8A, arrowheads). While we did not evaluate heart function or mortality, reovirus infection can cause lethal myocarditis at this dose at later time points [48]. Moreover, severe myocarditis impairs ventricular function, which can result in edema and weight gain. Not surprisingly, WT- and R296T-inoculated mice tended towards higher peak and mean weights 8 d pi than mice inoculated with K287T virus (S13 Fig). To directly quantify myocarditic damage in an independent cohort of mice, we processed and stained reovirus-infected heart tissue with Alizarin red to detect calcium deposits and reovirus antiserum to detect virus-infected cells (Fig 8B). We observed decreased foci of both viral replication and myocardial damage in the K287T-inoculated mice at 8 d pi (Fig 8B and 8C).

Based on these results, we speculated that K287T infection in mice initiates SG formation and an antiviral response early in infection that impairs late steps in disease pathogenesis. To test this hypothesis, we conducted RNA sequencing (RNA-seq) of reovirus-infected heart tissue at 4 dpi to identify early changes in host transcripts that may lead to the myocarditic

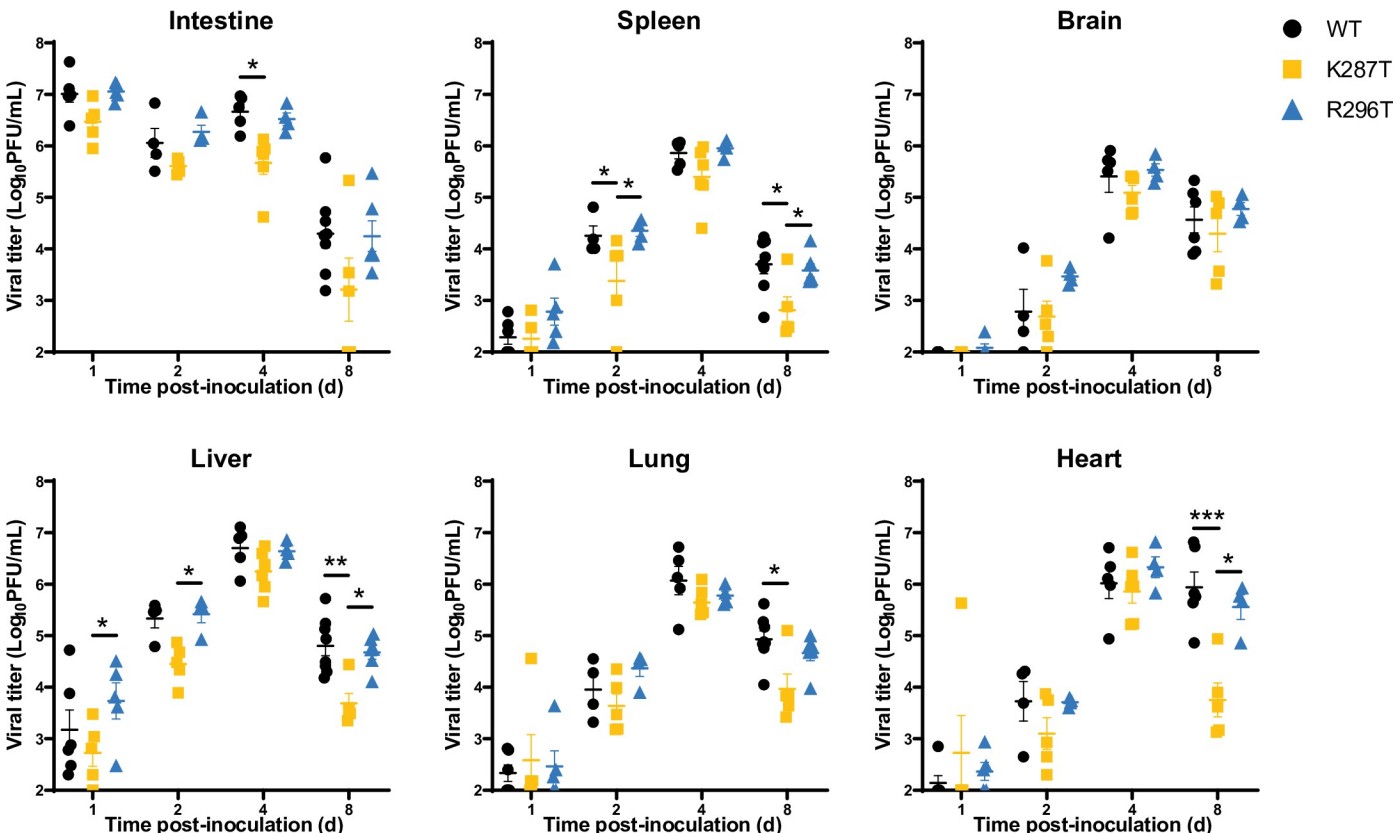

**Fig 7. T1L-K287T displays tissue-specific impairment of viral replication.** C57BL/6J 3-to-4-day-old mice were inoculated perorally with $10^7$ PFU of WT, K287T, or R296T virus. At the indicated times post-inoculation, mice were euthanized, and viral titers in organ homogenates were determined by plaque assay. Each symbol represents the viral titer of one mouse ($n$ = 4–8 mice/group). Data are log transformed and plotted on a linear scale. Mean viral titer is shown, and error bars indicate SEM. $P$ values were calculated using a two-way analysis of variance (ANOVA) followed by Tukey's test (*, $P < 0.05$; **, $P < 0.01$; ***, $P < 0.001$).

phenotype observed at 8 dpi. RNA was purified from hearts of mock-, WT-, K287T-, R296T-infected mice (3 mice per group) and subjected to sequencing. Transcripts were quantified and compared between mock-infected and virus-infected mice. The differentially expressed genes (DEGs) with a false discovery rate (FDR) $p$-value of less than 0.05 and an absolute $\log_2$ fold change of greater than 1.5 were identified (Fig 9A). A total of 268, 227, and 291 upregulated genes were detected in WT-, K287T-, and R296T-infected hearts, respectively, while only a few downregulated genes were identified in all three groups (Fig 9B and S3 Table). As expected, immune and antiviral associated genes were upregulated. Of the upregulated DEGs identified, 212 are shared between WT and both mutant viruses. Only 18 genes for WT, 11 genes for K287T, and 41 genes for R296T were uniquely upregulated in each group. Collectively, there were no significant changes in gene expression between WT and either mutant virus. However, when we compared the $\log_2$ fold changes of the 212 shared upregulated genes between WT and K287T or between WT and 296T ($\log_2$ fold change [WT]—$\log_2$ fold change [mutant]), we found that the shared genes were not as abundantly upregulated in K287T-infected mice. Indeed, the structure of volcano plots (Fig 9A) shows a similar trend that genes in the K287T-infected mice were upregulated to a lesser extent than those in WT- and R296T-infected mice. In R296T-inoculated hearts, these genes are upregulated to the same or sometimes greater extent than in WT-infected hearts (Fig 9C). When comparing the expression levels of these 212 DEGs between WT and each mutant, we found no statistically significant

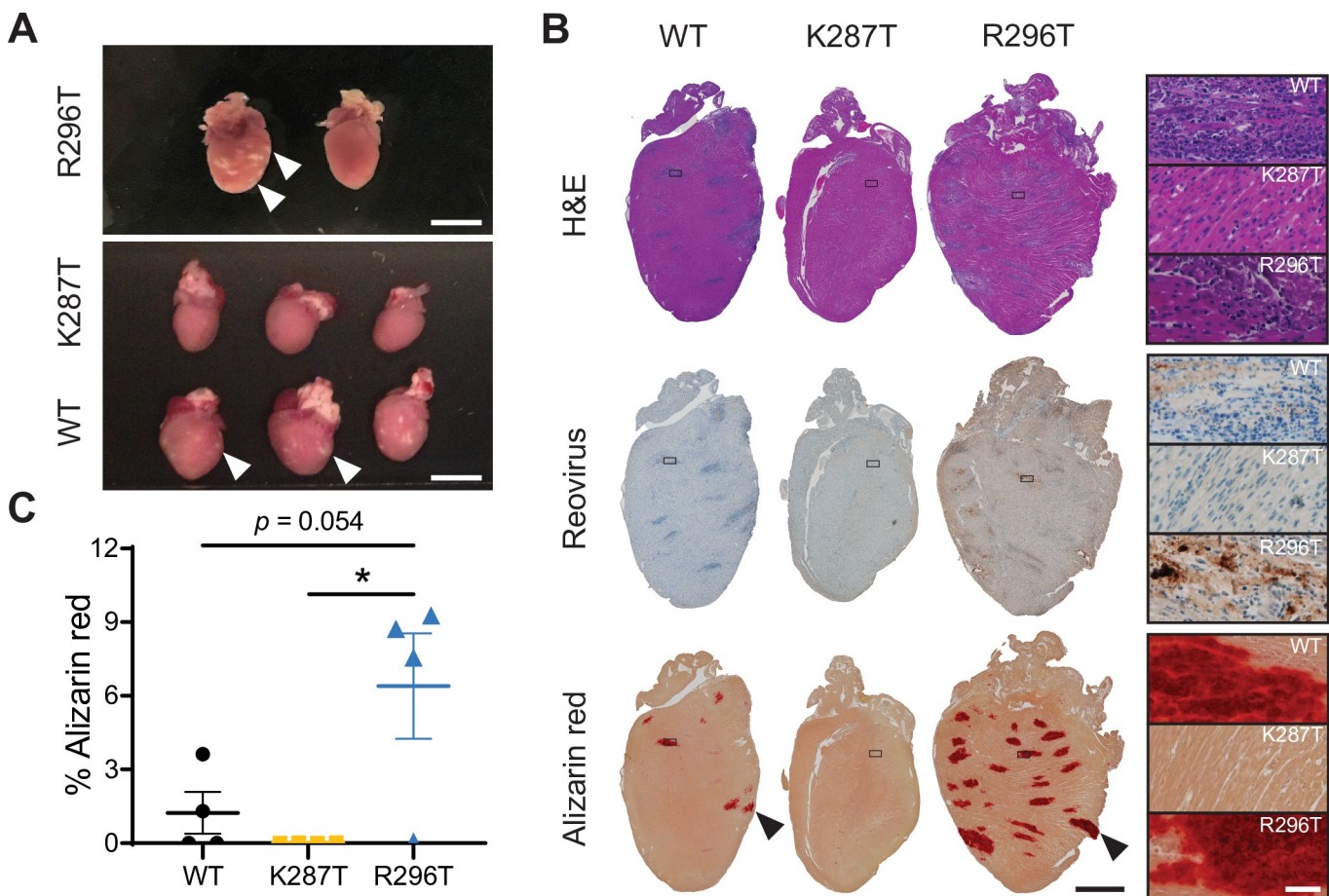

**Fig 8. Mice inoculated with T1L-K287T do not develop myocarditis.** (A-C) C57BL/6J 3-4-day-old mice were inoculated perorally with $10^7$ PFU of WT, K287T, or R296T virus. Mice were euthanized 8 d pi. Hearts were photographed, processed for histochemistry, sectioned, and stained with either hematoxylin and eosin (H&E), reovirus polyclonal antiserum, or alizarin red. (A) Gross pathology of hearts. Arrowheads indicate myocarditic lesions visible as white defects in cardiac tissue. Scale bar, 5 mm. (B) Representative images of heart sections for the indicated viruses and stains are shown, with inset regions indicated by a box. Enlarged insets are shown. Scale bar for left panels, 2000 μm. Scale bar for insets, 50 μm. (C) Myocarditis severity was scored by quantifying the percent area of heart tissue stained with alizarin red. Each symbol indicates one mouse. *P* values were calculated using a one-way ANOVA followed by Tukey's test (*, $P < 0.05$).

difference across any gene. Thus, a dsRNA-binding mutant of σ3 that promotes activation of PKR and abundant SG formation *in vitro*, displays tissue-specific attenuation in viral replication and disease in mice, likely due to activation of a coordinated innate immune response to dampen viral infection.

## Discussion

During viral replication, many if not all viruses synthesize dsRNA, which is a potent pathogen-associated molecular pattern sensed in the cytoplasm of cells by pattern-recognition receptors (PRRs) such as RIG-I and MDA5 [3]. PRR-activated signaling induces secretion of type I and III IFNs and, on binding of these cytokines to their cognate receptors, an antiviral response is evoked through the synthesis of IFN-stimulated gene products. Many viruses encode dsRNA-binding proteins that may act to bind and sequester dsRNA to prevent activation of PRRs such as RIG-I and MDA5, effector kinases such as PKR, and other dsRNA-activated host defense pathways (e.g., 2′, 5′-oligoadenylate synthase-RNase L) [49]. Although this is an appealing

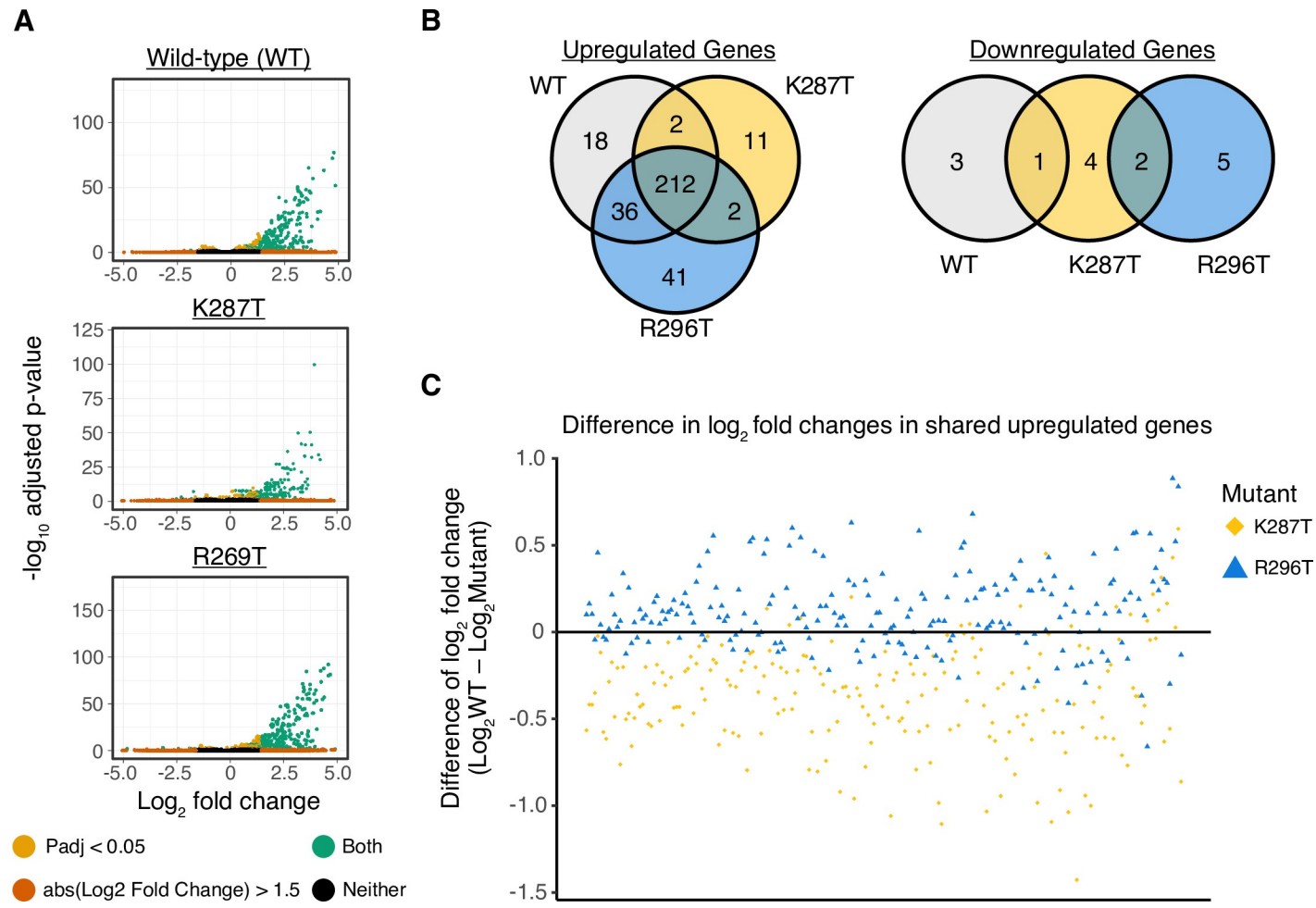

**Fig 9. T1L-K287T and T1L-R296T viruses exert minor differences on overall transcriptional responses in mouse hearts 4 d pi.** (A) Volcano plots depicting relative fold changes of individual genes in reovirus infected (WT, mutants K287T and R296T) mice as compared to uninfected mice. Genes are colored based on an absolute log$_2$ fold change greater than 1.5 (orange), an adjusted p-value less than 0.05 (gold), or both (green). (B) Venn diagrams illustrating the number of unique and shared differentially regulated genes (log2 fold change greater or less than 1.5, adjusted *p*-value < 0.05) across each reovirus infected group. (C) Difference of log2 fold changes across the 212 shared upregulated genes as compared to wild-type expression (log2 fold change (WT)-log2 fold change (mutant)). Points that are plotted on the horizontal line marked at 0 indicate no difference in fold change. Points plotted above the line indicate higher expression than wild-type and those plotted below the line indicate lower expression than wild-type. No statistical significance in differential gene expression (mutant vs wild-type) was observed across any of these 212 genes.

model, supporting evidence for some viruses has been inconsistent, and the dsRNA-binding capacity of these proteins may not function to sequester dsRNA. For example, the capacity of the vaccinia virus (VV) E3L protein to inhibit PKR activation does not correlate with its capacity to bind dsRNA [50]. Furthermore, activation of PKR by VV lacking E3L was not rescued by expression of a dsRNA-binding ortholog from sheep pox virus but was complemented by a myxoma virus orthologue [51]. Similarly, the influenza dsRNA-binding protein NS1 inhibits PKR activation by a direct interaction that is thought to prevent the dsRNA-induced conformational change in the protein rather than by sequestering dsRNA. Indeed, a recombinant influenza virus bearing a dsRNA-binding defective NS1 still blocks activation of PKR [52].

The mammalian orthoreovirus dsRNA-binding protein σ3 suppresses dsRNA-mediated activation of PKR [38]. It was hypothesized that σ3 binds and sequesters free dsRNA to prevent activation of PKR and subsequent inhibition of global translation [17,34,35,38,53,54]. Support

for this hypothesis was provided by *in vitro* findings that addition of purified σ3 blocks activation of partially purified PKR and the identification of three σ3 residues important for dsRNA-binding that fail to prevent activation of PKR [17]. Although these findings are compelling, they only correlate the dsRNA-binding capacity of σ3 with inhibition of PKR activation. Importantly, the function of dsRNA-binding by σ3 in the context of reovirus infection has not been examined. Here, we found that six basic residues (R202, R208, K287, K293, R296, and R326) that lie across the dimer interface of σ3 in a patch are each involved in binding dsRNA (Fig 1), as was predicted from the structure of σ3 [33]. We found that residues previously reported as being involved in binding dsRNA (K291, R239, and R236) were poorly expressed when they were exchanged with threonine residues (Fig 1). We noted that these residues are fully or partially buried in the structure of the σ3 dimer [33], suggesting that alterations of those residues cause σ3 to misfold. We focused on two mutants, K287T and R296T, which displayed no detectable dsRNA-binding and were recovered as recombinant viruses in the T1L background.

Although K287T and R296T were both defective in dsRNA-binding, only the K287T virus had a replication defect in A549 cells relative to WT virus (Fig 2). We found that the K287T mutant was incapable of suppressing activation of PKR or phosphorylation of eIF2α during infection, whereas the R296T mutant suppressed PKR phosphorylation and eIF2α phosphorylation comparable to WT virus (Figs 3 and 6). We also found that the K287T mutant produced decreased levels of viral proteins at 18 h pi. We attribute the decreased protein expression to failure of σ3-K287T to suppress PKR-mediated phosphorylation of eIF2α, as the replication defect of K287T as well as protein expression was rescued in PKR KO A549 cells (Figs 3 and 6). Interestingly, despite lower levels of protein expression, viral mRNA levels were higher in K287T-infected cells than in cells infected with WT virus, and viral mRNA could be detected in SGs (Figs 3 and 5). One possible explanation for these findings is that viral mRNA, possibly associated with stalled 48S ribosomes, is sequestered in SGs with a concomitant decrease in viral mRNA turnover. Additionally, the increase in viral mRNA could be due to decreased efficiency of assortment and packaging of mRNA into viral core particles. Consistent with this possibility, a role for σ3 in viral mRNA assortment has been proposed [55]. It also is possible that the overall decrease in viral protein synthesis leads to lack of outer-capsid proteins for assembly onto viral core particles leading to prolonged core transcription in K287T-infected cells. Collectively, these findings indicate that (i) inhibition of dsRNA-mediated activation of PKR by σ3 does not require dsRNA binding, (ii) efficient viral replication requires PKR suppression by σ3, and (iii) σ3 is required for suppressing SG formation.

The phenotypic difference between the K287T and R296T mutants despite both being incapable of binding dsRNA raises three questions. What is the mechanism of PKR inhibition? Why do these mutants differ in this capacity? And what is the biological function, if any, of the dsRNA-binding capacity of WT σ3? Other viruses inhibit PKR by a variety of mechanisms and many do so by direct interactions with PKR [56]. For example, influenza NS1 binds directly to the linker region of PKR between its N-terminal RNA-binding domain and C-terminal effector domain, preventing autophosphorylation of PKR upon binding of dsRNA [57]. In addition, NS1 binds to NF90, a cellular protein that acts to promote PKR phosphorylation preventing NF90 from activating PKR [58,59]. Such protein-protein interactions do not require the dsRNA-binding capacity of NS1, and we speculate that σ3 may function similarly. A specific protein-protein interaction between σ3 and PKR or other host proteins involved in the regulation of PKR activity might explain why the mutants differ in phenotype. Our ongoing studies are testing this hypothesis. Clearly, given the near WT replication of the R296T mutant, the dsRNA-binding function of σ3 is not required for viral replication in vitro or in vivo.

The presence of abundant G3BP1-positive, TIA-1-positive SGs was a prominent feature in A549 cells infected with the K287T mutant virus (Figs 6 and S8). SGs are liquid-liquid phase-separated organelles that form in response to a variety of environmental stresses (e.g., virus infection, oxidative stress, heat shock, cold shock, osmotic stress, and UV irradiation) in a process dependent on disassembly of polysomes [60]. SGs appear to form as a result of increasing concentrations of non-protein-bound or free RNA in the cytoplasm, as would occur after polysome disassembly [61,62]. Canonical SGs contain multiple RNA-binding proteins, components of the cellular translational machinery, and mRNAs associated with stalled 48S ribosomes [8]. SGs form in reovirus-infected cells early after infection (2 to 6 h pi) but then largely dissipate by ~ 12 h pi [28]. Later in the infectious cycle (24 h pi), infected cells are resistant to SA-induced formation of SGs [29]. The early induction of SGs by reovirus infection requires uncoating of incoming viral particles, but viral transcription or translation of viral proteins is not required [28]. As phosphorylation of eIF2α is required for early SG formation following reovirus infection [28], it is likely that release of dsRNA from incoming damaged particles activates PKR leading to phosphorylation of eIF2α, disassembly of polysomes, and assembly of SGs. Damaged core particles may enter the cytoplasm, or dsRNA may be released from degraded core particles in endolysosomes and then access the cytoplasm by RNA transmembrane transporters such as SIDT1 or SIDT2 [63,64]. While this seems likely, early SG formation still occurs in PKR KO cells and in cells lacking each of the three other individual kinases responsible for phosphorylation of eIF2α (HRI, PERK, and GCN2), suggesting that more than one stress kinase was activated early in reovirus infection [28]. We found that transfection of poly I:C (a dsRNA mimic) or infection with the R296T mutant induced the formation of G3BP1-positive/TIA1-negative RNase L-dependent bodies (RLBs) in WT cells and PKR KO cells, whereas infection with the K287T mutant induced RLBs in PKR KO cells (S10 Fig). RLBs form in WT A549 cells treated with poly I:C (S9 Fig) and, although RLBs share much of the protein content of SGs, they are distinct in that they do not contain TIA1 [13]. RNP granules are formed in WT, PKR KO, and RNase L KO cells infected with K287T or R296T (S10 Fig). However, we did not detect RNP granules in A549 cells lacking both PKR and RNase L following infection with WT or mutant viruses at 18 h pi. We hypothesize that the granules formed in PKR KO cells are RLBs rather than SGs [28]. Based on our observations and previously published results, we propose that the RNP granules that form in reovirus-infected cells between 2 and 6 h after infection occur secondary to cytoplasmic release of dsRNA from incoming particles that subsequently activates PKR or the OAS/RNase L signaling pathways.

At late times post-infection (24 h pi), SGs are absent from reovirus-infected cells, and they cannot be induced by treatment with SA [29]. The late suppression of SGs occurs independently of eIF2α phosphorylation, and drugs that inhibit translational initiation by binding to eIF4α are also incapable of inducing SGs in reovirus-infected cells, indicating that the mechanism of late suppression of SGs does not involve suppression of PKR activation or phosphorylation of eIF2α. In contrast, SA treatment and other treatments that lead to polysome disassembly at 6 h pi can still induce SG formation [31]. Considered together with our current findings, the early appearance of SGs dependent on eIF2α phosphorylation and the late suppression of SGs independent of eIF2α phosphorylation during reovirus infection suggests that suppression of PKR phosphorylation by σ3 is important early in the infectious cycle. Paradoxically, the replication of some reovirus strains (e.g., T2J and T3D) is impaired in PKR KO and RNase L KO cells [20]. Although replication of reovirus T1L is moderately suppressed in PKR KO A549 cells compared with its replication in WT cells, there is little effect on replication in RNase L KO cells (Fig 6B). Similarly, replication of strains T2J and T3D is decreased by ~ 10-fold in MEFs expressing the S51A non-phosphorylatable form of eIF2α [21]. These findings

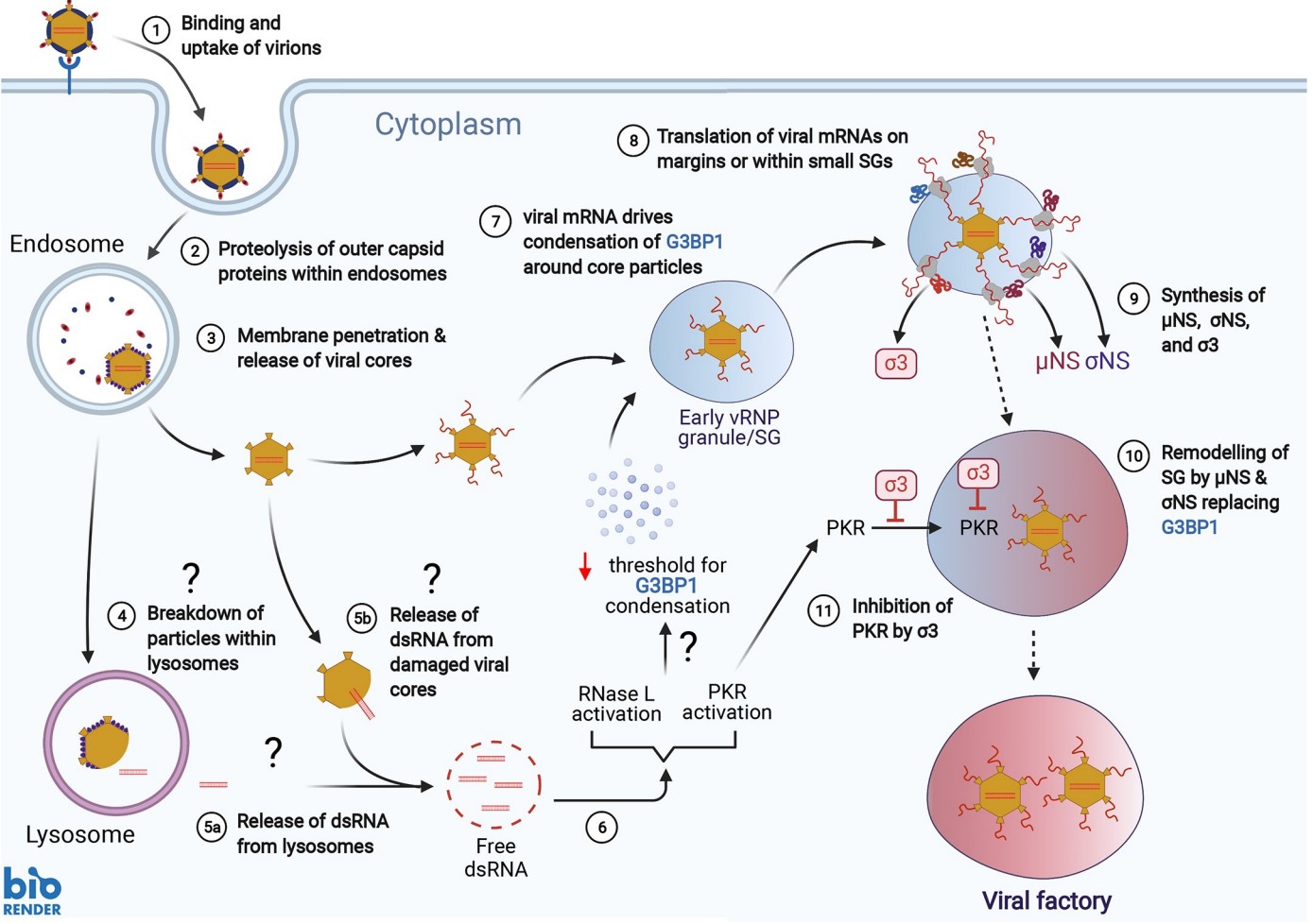

**Fig 10. Model for the function of σ3 in establishment of viral factories in reovirus-infected cells.**

suggest that at least some degree of phosphorylation of eIF2α is beneficial in some way for reovirus replication.

Based on our current findings and previously published results, we suggest a model for the early action of σ3 and the development of VFs (Fig 10). We propose that initial formation of RNP granules (SGs or RLBs) is induced by release of dsRNA from damaged incoming reovirus particles or from dsRNA that forms following folding of newly transcribed viral mRNAs. In support of the latter possibility, the reovirus s1 mRNA is a potent activator of PKR [65]. The presence of dsRNA in the cytoplasm then promotes the formation of SGs, RLBs, or both. G3BP1 appears to be required for the phase separation that leads to assembly of SGs and presumably RLBs. Post-translational modifications of G3BP1 alter its propensity to condense into SGs with demethylation of arginine residues in its RGG motif being associated with increased SG formation in response to SA-mediated phosphorylation of eIF2α [66]. In contrast, phosphorylation of G3BP1 at serine 149, which is located in the first of three intrinsically disordered regions in the protein, by casein kinase 2 increases its threshold for RNA-driven stress granule formation [61,62,67]. We hypothesize that changes in the posttranslational modification status of G3BP1 driven by signaling events following PKR phosphorylation or RNase L activation leads G3BP1 to undergo RNA-driven phase transition. Thus, early activation of

PKR or RNase L by dsRNA released during reovirus entry or formed on initial viral transcription would produce a more favorable cytosolic environment for RNA-driven condensation of G3BP1 into RNP granules. Subsequent transcription of viral mRNAs would release free RNA locally into the cytosol around core particles that would drive the formation of small phase-separated G3BP1-containing RNP granules. Small SGs do not contain phosphorylated eIF2α, as recruitment of PKR and phosphorylation of eIF2α occur after the SGs reach a critical size [68]. Viral mRNAs transcribed in small SGs would be retained by interactions with G3BP1 and other RNA-binding proteins and thus prevented from diffusing away from the viral core particle. We speculate that translation of early viral mRNAs can occur in small G3BP1-containing condensates that surround transcribing core particles, in keeping with observations that translation of some cellular proteins occurs in SGs [69]. We hypothesize that early formation of such granules around transcribing cores is beneficial for viral replication. This concept is supported by the observation that pre-treatment of cells with SA enhances reovirus replication and protein synthesis [30]. Synthesis of viral proteins in close proximity to the transcribing core particle would promote the formation of VFs around these particles. Importantly, PKR is activated following recruitment to SGs, but it is not recruited until SGs reach sufficient size [68,70,71]. Expression of σ3 would act to prevent activation of PKR in the enlarging SG/VF and allow viral mRNA translation to continue. As the concentration of reovirus nonstructural proteins σNS and μNS increases, we envision that VFs would replace the initial G3BP1-dependent SG replication site and act to suppress *de novo* SG formation. The reovirus nonstructural protein σNS associates with G3BP1, and co-expression of σNS with μNS suppresses SA-induction of SG formation [31]. In infected cells and cells expressing μNS and σNS, SG proteins such as G3BP1 become redistributed around the periphery of the VFs or μNS/σNS VF-like structures [31]. We suggest that the capacity of SGs to recruit RNA-binding proteins and components of the translational machinery is usurped by the nascent VFs. Others have noted the occasional co-localization between G3BP1 and other proteins that localize to SGs and μNS-containing VFs [29,31], and here we often noted co-localization between SG proteins G3BP1 and TIAR with μNS in K287T-infected cells (Fig 4C). Translation is compartmentalized in VFs in reovirus-infected cells, and active translation likely occurs on the margins of VFs [72]. Thus, we propose that reoviruses use SGs to seed the formation of VFs around transcribing viral core particles. Although virions of strain T1L used in the current study did not induce SGs in a large percentage of A549 cells at early times post-infection, in vivo, virions are converted to ISVPs by proteolytic enzymes in the gut prior to entry via M cells [73,74]. T1L ISVPs robustly induce SGs at early times post-infection in ~ 10–40% of cells, depending on the multiplicity of infection [28].

Infection by some reovirus strains causes myocarditis in neonatal mice [75]. Reovirus-induced myocarditis is not immune-mediated, and infection will cause myocarditic lesions in nude mice or mice lacking B- and T-cells [47,76]. Reovirus-induced cardiac damage occurs as a consequence virus-induced apoptosis of myocytes, and disease does not occur in mice lacking caspase 3 or in WT mice treated with inhibitors of apoptosis [77–80]. Comparisons of reovirus strains that vary in the capacity to cause myocarditis have identified phenotypic and genetic differences that correlate with myocarditic potential. Myocarditic strains synthesize more viral mRNA than non-myocarditic strains in infected primary cardiac myocytes, and this phenotype is determined by polymorphisms in the S1 and M1 genes responsible for encoding the σ1/σ1s and μ2 viral proteins, respectively. Despite synthesizing larger quantities of viral mRNA, myocarditic strains do not replicate to higher titers than non-myocarditic strains in primary cardiac myocytes, and myocarditic and non-myocarditic strains are equally efficient at infecting primary cardiac myocytes [81]. Increased synthesis of viral mRNA by myocarditic strains is a consequence of their increased capacity to suppress antiviral responses.

Myocarditic strains induce less IFN-α/β than non-myocarditic strains in infected myocytes or are less sensitive to the antiviral effects of IFNs [82]. Reovirus genes M1, L2, and S2 are determinants of viral strain-specific differences in induction and sensitivity to IFN-α/β [81]. However, the reovirus μ2-encoding M1 gene is the primary determinant of strain differences in reovirus-induced myocarditis. As a consequence of a single amino-acid polymorphism at amino acid residue 208, the μ2 protein from myocarditic strains can repress IFN-β signaling [83]. The mechanism by which μ2 represses IFN signaling is likely multifactorial. Inhibition of IFN signaling by μ2 correlates with nuclear distribution of IFN regulatory factor 9 (IRF9), and μ2 interacts with the mRNA splicing factor SRSF2 in nuclear speckles and modifies cellular splicing, perhaps diminishing the IFN response [84,85]. The idea that myocarditic strains that synthesize greater amounts of viral RNA in cardiac myocytes would synthesize less IFN is somewhat puzzling, as increased viral RNA could be thought to better stimulate PRRs and lead to an enhanced IFN response. One possible explanation is that the negative effects of increased viral RNA are counteracted by the increased translation and synthesis of σ3, which acts to dampen the effects of dsRNA and limit activation of PRRs and subsequent IFN synthesis [38,86]. The σ3 protein may play a direct role in suppression of IFN-β synthesis independent of its capacity to bind dsRNA [86]. The pathway of induction of type I IFN by reovirus infection in cardiac myocytes requires PKR, as synthesis of IFN is severely compromised in PKR KO cardiac myocytes [87]. Similarly, IFN regulatory factor 3 (IRF3) is required for induction of IFN-β in primary cardiac myocytes, and mice lacking IRF3 develop severe myocarditis [88,89]. Finally, mice lacking the p50 subunit of NF-κB are highly susceptible to reovirus-induced myocarditis such that normally non-myocarditic strains can cause the disease [90]. It is thought that activation of NF-κB in cardiac myocytes is required for an IFN-α/β response to viral infection. However, activation of NF-κB in primary cardiac myocytes is largely recalcitrant to reovirus infection or treatment with tumor necrosis factor-α, demonstrating only a minimal response relative to cardiac fibroblasts. Primary cardiac myocytes express high basal levels of type I IFNs because of constitutive activation of the mitochondrial antiviral signaling pathway (MAVS) [91], and it is possible that the minimal NF-κB response to infection is sufficient to stimulate a protective IFN response. The minimal NF-κB response may benefit cardiac myocytes by preventing pro-apoptotic effects of NF-κB [92]. Indeed, NF-κB is not required for reovirus-induced apoptosis in the heart [90]. Our findings that the σ3-K287T dsRNA-binding mutant virus does not induce myocarditis in mice and activates PKR and phosphorylation of eIF2α fit with previous observations of the protective role of PKR in suppressing reovirus-induced myocarditis [87]. The other σ3 dsRNA-binding mutant virus (R296T) did not activate PKR in cultured cells and induced myocarditis in mice. Similar to WT T1L, the R296T mutant is capable of suppressing an IFN response in HEK293 cells [86], indicating that dsRNA-binding also is not required for this function.

We show that the reovirus σ3 protein is required for viral pathogenesis and functions to suppress PKR activation independently of its capacity to bind dsRNA. This work contributes new information about properties of σ3 and enhances an understanding of how viruses evade cell-intrinsic defenses to achieve efficient replication.

## Materials and methods

### Ethics statement

All animal husbandry and work using animals was conducted ethically, conform to the U.S. Public Health Service policy, and was approved by the Institutional Animal Care and Use Committee at the University of Pittsburgh.

## Cells and viruses

HEK293T were cultured in Dulbecco's modified Eagle's medium (DMEM) supplemented to contain 10% FBS, 1% L-glutamine, 1% sodium pyruvate, and 1% penicillin-streptomycin. Caco-2 cells were cultured in EMEM supplemented to contain 20% FBS, 1% L-glutamine, 1% non-essential amino acids, and 1% penicillin-streptomycin. L929 cells were cultured in Joklik's modified Eagle's minimal essential medium (Sigma) supplemented to contain 5% FBS, 2 mM L-glutamine, 1% penicillin-streptomycin and 0.25 μg/ml of amphotericin B. BHK-T7 cells were cultured in DMEM supplemented to contain 10% FBS, 1% L-glutamine, 1% penicillin-streptomycin and 1 mg/ml G418. A549 parental cells, PKR KO, RNase L KO, PKR/RNase L DKO cells were provided by Dr. Roy Parker's lab at the University of Colorado and cultured in DMEM/F12 supplemented to contain 10% FBS, 1% L-glutamine, and 1% penicillin-streptomycin.

All viruses used in this paper were recovered using reverse genetics [93,94]. Reovirus gene segments in pT7 vector plasmids were co-transfected with pCAG-FAST, pCAG-D1R, and pCAG-D12L into BHK-T7 cells using TransIT-LT1 (Mirus) following the manufacturer's instructions. Cells were incubated at 37˚C for 2 d before being frozen at -80˚C. Two freeze-thaw cycles were performed before plaque isolation. All viruses were purified, and dsRNA genome material was extracted and verified by Sanger sequencing.

## Plasmids

For engineering of plasmids that express σ3 WT and mutants, σ3 WT and mutant sequences flanked by BamHI and EcoRI sites were synthesized commercially by ATUM and then cloned into pWPXL-GFP plasmids to replace GFP sequence through T4 ligation (NEB). pWPXL-Flag and pWPXL-HA vectors were generated by ATUM by replacing GFP with Flag- or HA-'stuffer sequences' flanked by BamHI and EcoRI sites. Then, σ3 WT or mutant sequences were cloned into pWPXL-Flag or pWPXL-HA using T4 ligation. For pT7-σ3(T1L) or pT7-σ3(T3D) plasmids, small DNA fragments (~ 200 bp) containing the indicated σ3 mutant sites were synthesized by ATUM and were used to replace the pT7- σ3 WT sequences using T4 ligation.

To establish the doxycycline inducible σ3-expressing stable cells lines, we engineered the lentiviral vectors pCW57.1-σ3-WT, pCW57.1-σ3-K287T and pCW57.1-σ3-R296T by Gateway cloning (Thermo Fisher Scientific). cDNAs encoding σ3 flanked by attB sequences were amplified from pT7-σ3 constructs and subcloned into the pDONR-201 vector (provided by Dr. Michael S. Y. Huen) by BP reaction. An LR reaction was then conducted to subclone σ3 into the pCW57.1 vector (provided by Dr. David Root; Addgene plasmid # 41393). All σ3-encoding sequences, either WT or mutants, were confirmed by Sanger sequencing.

## Production of lentiviral particles

To produce σ3-expressing lentiviral particles, HEK293T cells (10 cm$^2$ dish) were co-transfected with 6 μg of pCW57.1-σ3-WT, pCW57.1-σ3-K287T, or pCW57.1-σ3-R296T, together with 3 μg of psPAX2 (Addgene plasmids #12260) and 1.5 μg of pMD2.G (Addgene plasmid #12259) using 31.5 μl of *Trans*IT-Lenti Transfection reagent (Mirus). The medium was collected 24 h post-transfection and 48 h post-transfection. Lentiviral particles were concentrated using PEG-it virus precipitation solution (System Biosciences) following the manufacturer's instructions and resuspended in phosphate-buffered saline (PBS).

## Establishment of stable cell lines

To establish A549 or A549 RNase L KO stable cell lines that express σ3, A549 or A549 RNase L KO cells were resuspended in medium containing 200 μl lentiviral particles and 8 μl/ml

polybrene. Cells were seeded into T75 flasks. After 24 h post-transduction, cells were trypsinized and seeded back into selective growth medium containing 10 μg/ml puromycin (Sigma-Aldrich). Selective medium was changed every three days. Single colonies were picked after serial dilution and verified by immunoblotting and immunofluorescence for σ3 expression following doxycycline treatment.

### SDS-PAGE, immunoblotting, and antibodies

To detect σ3, μ1, μNS, σNS, actin, and other unphosphorylated cellular proteins, cell pellets were lysed on ice in NETN buffer (100 mM NaCl; 20 mM Tris-Cl pH 8.0; 0.5 mM EDTA; 0.5% [v/v] Nonidet P-40) for 30 min before being diluted in 5X SDS-PAGE sample buffer. Samples were then boiled at 95˚C for 10 min and resolved in 4–15% Mini-PROTEAN TGX Precast Protein Gels (BioRad) for immunoblot analysis. To detect phosphorylated PKR and eIF2a, cell pellets were collected in denaturing lysis buffer (20 mM Tris-Cl pH 8.0; 50 mM NaCl; 0.5% [v/v] Nonidet P-40; 0.5% deoxycholate; 0.5% SDS; 1 mM EDTA). After 30 min incubation on ice, lysate was then diluted in 5X SDS-PAGE sample buffer and boiled at 95˚C for 10 min prior to immunoblotting analysis.

The following antibodies were used for immunoblotting: chicken anti-μNS polyclonal antibody [72], guinea pig anti-σNS polyclonal antibody [95], and rabbit anti-σ3 polyclonal antibody [96]. The mouse anti-σ3 monoclonal antibody 4F2, mouse anti-μ1 antibody 4A3, and mouse anti-λ2 antibody 7F4 were deposited to the DSHB by Dr. Terence Dermody. Other commercial antibodies used are detailed in S1 Table.

### Puromycin labeling

Cells were treated with poly I:C or infected with various virus strains. At the time of labeling, cells were either untreated or treated with 208 μM emetine at 37˚C for 15 min and incubated with 182 μM puromycin at 37˚C for 5 min. Following incubation, cells were immediately placed on ice and washed twice with ice-cold PBS. After washing, cells were lysed using NETN buffer and electrophoresed and immunoblotted for the presence of puromycylated adducts.

### Poly (I:C) pull down assay

Poly (C) coated agarose beads were resuspended with poly (I) in 50 mM Tris-Cl pH 7.0–150 mM NaCl to form poly (I:C) beads as previously described [97,98]. Poly (I:C) Beads were then incubated with cell lysates derived from pWPXL-σ3 transfected cells. Two h post incubation, beads were pelleted by centrifugation at 1700 rpm for 2 min and washed 4 times using NETN buffer. Beads were boiled in SDS sample buffer. Eluted proteins were analyzed by immunoblotting.

### dsRNA extraction from reovirus-infected cells

To verify sequences of recombinant viruses, dsRNA was extracted from reovirus-infected cells for Sanger sequencing. Generally, L929 cells were infected with P2 virus stock at 5 PFU per cell. Cells were harvested for total RNA extraction using TRIzol reagent (ThermoFisher) when > 30% of cells showed cytopathic effect. After total RNA extraction, lithium chloride was added (final concentration is 2 M) to precipitate ssRNA at 4˚C for at least 8 h. RNA was pelleted by centrifugation in a microcentrifuge at 12,000 rpm for 10 min. The supernatant was transferred to a new tube and 400 μl of 100% ethanol and 33.33 μl of ammonium acetate (stock concentration 7.5 M) was added. After incubation at 4˚C for 2 h, dsRNA was pelleted, washed with 75% ethanol and resuspended in water. Viral dsRNA was denatured at 95˚C for 3 min

and then S4 gene segments were amplified by RT-PCR. PCR product was sent for Sanger sequencing.

### *In vitro* viral dsRNA pulldown

T1L viral dsRNA was extracted from purified T1L virions by TRIzol LS reagent according to the manufacturer's instructions (Thermo Fisher # 10296010). Viral dsRNA was conjugated with EZ-Link Psoralen-PEG3-Biotin (Thermo Fisher # 29986) by UV irradiation (365 nm) on ice for 30 min as described [99]. Excess biotin was removed by precipitation of the biotinylated-dsRNA using 0.3 M potassium acetate and 2.5 volume of ethanol. Biotinylated-dsRNA was added to cell lysates expressing various proteins and incubated at 4˚C for 2 h. Pierce Streptavidin Agarose (Thermo Fisher # 20353) was added into the mixtures and incubated at 4˚C for another 2 h. Agarose was washed five times with lysis buffer and immunoblotted to detect the dsRNA-binding capacity of the various proteins.

### Bone marrow-derived dendritic cells

Single cell suspensions of mouse bone marrow cells were cultured in RPMI 1640 medium supplemented with 10% FBS, 1% L-glutamine, 1% penicillin-streptomycin, and 20 ng/ml GM-CSF (PeproTech 315–03). Fresh medium containing 20 ng/ml GM-CSF was provided on d 3, d 6, and d 8 post-isolation. On d 10, cells were gently harvested and counted/validated using flow cytometry to detect surface markers CD11c+, MHC II, CD40, and CD80. Cells were then incubated with 10 PFU per cell of reovirus at 37˚C for 1 h. After 1 h of binding, cells were washed with PBS that was pre-warmed at 37˚C, resuspended in pre-warmed RPMI medium supplemented to contain 5 ng/ml GM-CSF, and seeded into non-tissue-culture treated 24-well plates at a density of $1\times10^6$ cells per well. At 24 h pi, cells were harvested to assess viral titer by plaque assay.

### Virus replication curves

L929 cells were inoculated with different multiplicities of recombinant reovirus (MOI 0.1 for multiple steps grow curve and MOI 5 for single step growth curve). At different intervals post-inoculation, viral titers were determined by plaque assay. For A549, Caco-2, and BMDCs, viruses were inoculated at 10 PFU per cell, and virus titer was measured 24 h pi.

### Co-immunoprecipitation assay

For dimerization assays, both Flag-tagged σ3 and HA-tagged σ3 were co-transfected into HEK293 cells at a 1:1 ratio. At 48 h post-transfection, cells were collected and lysed with NETN buffer on ice for 30 min. Lysates were cleared by centrifugation and supernatants were incubated with anti-Flag M2 agarose (Sigma) at 4˚C for 2 h. Beads were washed 4 times with NETN buffer and subsequently boiled with SDS sample buffer. HA was detected by immunoblot analysis.

For μ1-σ3 interaction studies, pCI-μ1 and pWPXL-σ3 were co-transfected into HEK293 cells. Cell lysates were incubated with μ1 monoclonal antibody 4A3 at 4˚C overnight, followed by incubation with recombinant Protein G agarose (Invitrogen) for 2 h at 4˚C. Beads were pelleted, washed 4 times with NETN, and boiled with SDS sample buffer.

### Immunofluorescence

Cells were seeded onto coverslips 24 h prior to infection. At 18 h after reovirus infection, or 8 h after poly I:C treatment, or 1 h after sodium arsenite (SA) treatment, cells were fixed in 3%

paraformaldehyde (PFA) in PBS at room temperature for 10 min. Cells were then incubated in permeabilization buffer (1% BSA and 0.1% TritonX-100 in PBS) for 15 min, followed by primary antibody incubation (diluted in PBS) at room temperature for 30 min and fluorescence-conjugated secondary antibody incubation at room temperature for 30 min. After washing, coverslips were then mounted onto glass slides with ProLong Gold Anti-Fade reagent with DAPI (Invitrogen). Images were collected by Nikon TE2000 inverted microscope and were processed using Photoshop (Adobe) and Illustrator (Adobe) software.

### Single-molecule fluorescence *in situ* hybridization (smFISH)

To visualize s4 mRNA in reovirus-infected cells, a set of s4 mRNA FISH probe consisting of 39 CAL Fluor Red 610 Dye-conjugated primers was designed using Stellaris Probe Designer (S2 Table). To co-stain for TIAR and λ2, a sequential IF +FISH protocol was used. Briefly, A549 cells were infected with 100 PFU per cell of reovirus and fixed with 3% PFA at 18 h pi. After incubation at room temperature for 10 min, cells were washed twice with PBS, incubated with permeabilization buffer for 15 min, followed by primary antibodies incubation that are specifically against TIAR and λ2 (diluted in PBS) at room temperature for 30 min, and fluorescence-conjugated secondary antibody incubation at room temperature for 30 min. After washing with PBS, cells were washed with 1 ml of Wash buffer A (Biosearch Technologies, SMF-WA1-60). To probe for s4 mRNA, cells were then incubated with s4 probes diluted in hybridization buffer (Biosearch Technologies, SMF-HB1-10) in a humidified chamber at 37˚C for 4 h in the dark. After that, cells were then washed in Wash buffer A at 37˚C for 1 h in dark. A final wash step was performed using wash buffer B (Biosearch Technologies Cat# SMF-WB1-2) for 5 min before mounting coverslips with ProLong Gold Anti-Fade reagent (Invitrogen). Images were collected using Olympus FLUOVIEW FV3000 and were processed using Photoshop (Adobe) and Illustrator (Adobe) software.

### Limited proteolysis of σ3 with trypsin

T1L WT or T1L σ3 mutant virions ($2\times10^{12}$ particles per ml in 100 μl) were incubated with 10 μg/ml trypsin (Sigma) at 8˚C in a MyCycler thermal cycler (Bio-Rad). At the indicated intervals, 10 μl of each reaction was mixed with SDS sample buffer and boiled for 10 min. Samples were analyzed by SDS-PAGE and Coomassie Brilliant Blue was used to stain gels. Band intensity was analyzed using Image J software.

### Infection of mice

Three-to-four-day-old C57BL/6J mice were inoculated perorally with $10^7$ PFU of purified reovirus diluted in PBS. Intramedic polyethylene 0.28/0.61 mm tubing, 1 mL slip-tip syringes, and a 30-gauge needle were used for inoculations. At various intervals post-inoculation, mice were euthanized by isoflurane with secondary decapitation, and tissues were collected.

For analysis of viral replication, organs were harvested into 1 mL of PBS, frozen and thawed twice, and homogenized using 5 mm stainless-steel beads and a TissueLyser (Qiagen). Virus titers in serum and organ homogenates were determined by plaque assay using L929 cells [100].

For immunohistological and RNA analyses, a separate cohort of similarly inoculated mice was euthanized at 8 or 4 dpi, respectively. Hearts were bisected, and half was collected into TRIzol and snap-frozen. Remaining halved-hearts were fixed in 10% formalin for 24–48 h and transferred to 70% EtOH prior to processing and embedding in paraffin. Consecutive 3-μm sections were stained with H&E, alizarin red, or reovirus antiserum, followed by a DAB secondary stain and hematoxylin counterstain.

All animal husbandry and work conform to the U.S. Public Health Service policy and was approved by the Institutional Animal Care and Use Committee at the University of Pittsburgh.

### Library preparation and RNA-seq analysis

Total RNA extracted from hearts 4 d pi (as described above) was used as input for library preparation and analysis. Experimental groups consisted of mock infected mice, T1L-WT, T1L-K287T, an T1L-R296T infected mice. For each experimental group, triplicate samples were obtained. Library preparation was prepared by Cornell University's Transcriptional Regulation and Expression Facility (TREx). Total RNA was first depleted of host (mouse) ribosomal RNA using the NEBNext rRNA Depletion kit according to manufacturer's protocol. Library preparation was performed using the NEBNext Ultra II RNA Library Prep kit according to manufacturer's protocol. Paired end sequencing of 150 base pairs per read was performed on an Illumina HiSeq 4000 instrument.

RNAseq datasets underwent quality control filtering and adapter removal using cutadapt version 2.1 [101] with the following settings: -a AGATCGGAAGAGC -A AGATCGGAAGA GC -g GCTCTTCCGATCT -G GCTCTTCCGATCT -a AGATGTGTATAAGAGACAG -A AGATGTGTATAAGAGACAG -g CTGTCTCTTATACACATCT -G CTGTCTCTTATACA CATCT, -q 30,30,—minimum-length 80, and–u 1. Remaining reads were mapped to the Mus musculus genome assembly GRCm38.95 from Ensembl using HISAT2 version 2.2.1 [102]. Read mapping was tabulated using featureCounts version 2.0.1 [103] to the Mus musculus GRCm38.95 gtf file with the following settings: -t exon -g gene_id. This read count table was used for differential gene expression analysis using DESeq2 version 1.22.2 [104] in Rstudio using R version 3.5.1. The raw sequencing datasets can be found on NCBI Sequence Read Archive (SRA) under BioProject PRJNA699030 and analysis scripts can be found at: https://github.com/scross92/sigma3_mutants_RNAseq_analysis.

### RT-qPCR

To quantify s4/m3 mRNA, total RNA extracted from reovirus-infected cells were used for reverse transcription using SuperScript III Reverse Transcriptase (Invitrogen) without pre-denaturing steps. Random hexamer primer (Thermo Fisher SO142) was used for RT steps. After cDNA was generated, s4/m3 mRNA or housekeeping gene *GAPDH* was assessed by Applied Biosystems Fast SYBR Green Master Mix (Thermo Fisher 43–856012). Primers sequences are as follows: s4 forward: 5′-CGCTTTTGAAGGTCGTGTATCA; s4 reverse: 5′-CT GGCTGTGCTGAGATTGTTTT; m3 forward: 5′-CGTGGTCATGGCTTCATTC; m3 reverse: 5′-CACATGCTGATAAGGTATAGACAT; GAPDH forward: 5′-TGATGACATCAAGAAG GTGGTGAAG; GAPDH reverse: 5′-TCCTTGGAGGCCATGTAGGCCAT).

To examine replication of viral genome, we used a modified qPCR assay to quantify s4 negative-sense ssRNA which was described previously [105]. Briefly, total RNA extracted from reovirus-infected cells was denatured at 95˚C for 3 min and subjected to reverse transcription using SuperScript III Reverse Transcriptase (Invitrogen). A primer (s4 forward: 5′-CGCTTTTGAAGGTCGTGTATCA) that only binds to the negative-sense strand of s4 was used to reverse transcribe s4 RNA to s4 negative-sense cDNA. Subsequent PCR was performed with both forward and reverse primers for s4 as described above. All results were normalized to T1L WT.

## Supporting information

**S1 Table. Commercial antibodies used in this study.**
(DOCX)

**S2 Table. List of oligonucleotide probes used for s4 mRNA single molecule FISH.**
(XLSX)

**S3 Table. List of genes differentially expressed in the hearts of mice infected with WT, K287T and R296T viruses at day 4 pi.**
(XLSX)

**S1 Data. All numerical data used for generation of figures in the manuscript.**
(XLSX)

**S1 Fig. Wildtype, but not K287T or R296T mutant σ3 can bind to biotinylated viral dsRNA.** Biotinylated viral dsRNA extracted from purified T1L virions was incubated with cell lysates expressing Flag-tagged σNS or σ3 WT or mutant proteins. Streptavidin-agarose pull-downs were washed extensively and immunoblots were performed to detect input expression levels and the viral dsRNA-binding capacity of the indicated proteins.
(TIF)

**S2 Fig. The capacity of mutants of σ3 to dimerize and co-assemble with μ1 is unaffected by their loss of dsRNA-binding activities.** (A) dsRNA-binding defective mutants of σ3 remain capable of dimerizing. HA-tagged σ3 was co-transfected together with Flag-tagged σ3 into HEK293 cells. At 48 h pi, coimmunoprecipitation of HA-tagged σ3 with FLAG-tagged σ3 WT or dsRNA-binding defective mutants from cell lysates was performed using anti-Flag agarose beads for 2 h, followed by western blotting to detect indicated proteins. (B) dsRNA-binding defective mutants of σ3 are capable of assembling with co-expressed μ1. μ1 was transfected into HEK293 cells together with either σ3 WT or dsRNA-binding defective mutants. At 48 h post-transfection, coimmunoprecipitation of WT and dsRNA-binding mutants of σ3 was performed using a monoclonal antibody against μ1(4A3). Unless specified, σ3 was detected by monoclonal antibody (4F2).
(TIF)

**S3 Fig.** Recombinant T3D and T1L viruses that carry dsRNA-binding defective σ3 mutations grow with similar kinetics to WT virus during (**A**) single and (**B**) multiple step replication in L929 cells. Cells were infected with the indicated WT or mutant viruses at 5 PFU per cell (single step replication) or 0.1 PFU per cell (multiple step replication). Change in viral titer was determined by plaque assay. Data are reported as mean ± S.D. of three independent experiments.
(TIF)

**S4 Fig. Levels of PKR and eIF2α phosphorylation in L929 cells infected with T1L-WT, T1L-K287T, and T1L-R296T are comparable.** (A) Cells were mock-infected or infected with the indicated viruses at 5 PFU per cell. At 18 h pi, cell lysates were collected in denaturing lysis buffer for immunoblotting and probed with antibodies against viral proteins μNS and σ3, and cellular proteins PKR, phosphorylated PKR (T446), phosphorylated eIF2α (S51), and actin. (B) Band intensities of phosphorylated PKR and phosphorylated eIF2α were quantified and normalized to total PKR and eIF2α respectively, and then normalized to mock (mock = 1). Data shown represent the mean ± s.d. of three independent experiments. Multiple comparison, non-paired t tests were used to analyze differences compared to WT (ns = not significant).
(TIF)

**S5 Fig. The *in vitro* trypsin sensitivity of σ3 on purified virions of T1L-WT, T1L-K287T and T1L-R296T are equivalent.** (A) Equal concentrations of purified virions were incubated

with 10 ng/ml trypsin at 8˚C. At the indicated time points, equivalent volumes of digestion aliquots were collected and analyzed by SDS-PAGE and Coomassie Brilliant Blue (representative experiment). (B) Band intensities of σ3 were measured using Image J software and normalized to the intensity at T = 0. Data shown represent the mean ± s.d. of two independent experiments.
(TIF)

**S6 Fig. T1L-K287T infection, but not T1L-WT or T1L-R296T, induces cellular translational shutoff.** (A) A549 cells were left untreated or treated with poly (I:C) at 1 μg/ml for 6 h prior to treatment with 208 μM emetine for 15 min at 37˚C followed by RPM labeling. Puromycin (PMY) incorporation levels were assessed by immunoblotting. (B) A549 cells were infected with T1L-WT, or T1L-K287T, or T1L-R296T at 100 PFU per cell. At 18 h pi, cells were treated with emetine for 15 min at 37˚C followed by RPM labeling. PMY incorporation were assessed by immunoblotting. (C) Lane intensities were measured using Image J software and normalized to mock (mock was set to 100). Data shown represent the mean ± s.d. of three independent experiments. Multiple comparison, non-paired t tests were used to analyze differences comparing with mock. (ns = not significant; *, $P < 0.05$).
(TIF)

**S7 Fig. s4 mRNA, but not GAPDH mRNA specifically localizes to viral factories.** A549 cells were either mock infected or infected with T1L-WT at 100 PFU per cell. At 18 h pi, cells were fixed for immunostaining with antibodies against viral protein λ2 followed by secondary antibodies staining. Subsequently, CAL Fluor Red 610 Dye-conjugated s4 mRNA probes or Quasar 670 Dye-conjugated GAPDH mRNA probes were used to detect s4 mRNA or GAPDH mRNA, respectively. Images were collected using Olympus FLUOVIEW FV3000.
(TIF)

**S8 Fig. Quantification of mean fluorescence levels of λ2 and s4 mRNA within cells infected with WT, K287T, and R296T viruses.** Colored circles represent individual infected cells categorized according to the distribution of s4 mRNA within each cell: A only (purple circles)—s4 mRNA co-localized within clearly defined VFs as detected by λ2 staining, SGs absent; B only–(yellow circles) s4 mRNA co-localized with TIAR in SGs; A and B–(blue circles)—a mixture of phenotypes A and B, SGs detected; A, B, and C–(green circles) a mixture of phenotypes A and B and in addition co-localization of s4 mRNA, TIAR, and λ2 within clearly defined VFs, SGs present. (A) Mean λ2 and (B) s4 mRNA fluorescence of individual A549 cells infected with WT, K287T and R296T viruses at 18 h pi. (C) Mean s4 mRNA fluorescence of K287T-infected cells grouped according to the presence or absence of SGs. (D) Mean λ2 and s4 mRNA fluorescence of all infected cells grouped according to the presence or absence of SGs. Multiple comparison, non-paired t tests were used to analyze differences, comparing with WT or no SGs, (ns = not significant; **, $P < 0.01$, ***, $P < 0.001$.
(TIF)

**S9 Fig. G3BP and TIAR localize to both RNase L-dependent bodies (RLBs) and SGs.** A549 cells or A549 RNase KO cells were transfected with poly I:C at 500 ng/ml using Lipofectamine 2000 following the manufacturer's instructions. At 8 h post-transfection, cells were fixed and immunostained for G3BP and TIAR. Nuclei were counterstained with DAPI.
(TIF)

**S10 Fig. Induction of RLBs and SGs in A549 WT, PKR KO, RNase KO and DKO cells by transfection with poly I:C or treatment with sodium arsenite.** Cells as indicated were transfected with 500 ng/ml poly I:C and then incubated for 8 h (upper) or treated with 0.5 mM

sodium arsenite for 1 h (middle) or left untreated (bottom). After transfection with poly I:C or treatment with SA, the cells were fixed and immunostained for TIA1 and G3BP. Nuclei were counterstained with DAPI.
(TIF)

**S11 Fig. Formation of SGs (TIA-1-positive, G3BP-positive) and RLBs (TIA-1-negative, G3BP-positive) in WT, PKR KO, RNase L KO, and DKO A549 cells infected with T1L-K287T and T1L-R296T viruses.** Cells as indicated were infected with either T1L-K287T or T1L-R296T virus at 100 PFU per cell. At 18 h pi, cells were fixed and immunostained for TIA1, G3BP and μNS. Merged image is showing colocalization of TIA1 and G3BP1. Images were collected using Olympus FLUOVIEW FV3000.
(TIF)

**S12 Fig. Ectopic σ3 expression can prevent poly I:C-induced PKR phosphorylation and the formation of SGs in RNase L KO cells, but is unable to prevent poly I:C-induced RLBs in A549 WT cells.** (A) Ectopic expression of WT σ3 or the R296T mutant, but not the K287T mutant prevents phosphorylation of PKR. σ3 protein expression was induced by treatment with doxycycline (1 μg/ml) for 24 h. Cells were then transfected with 500 ng/ml Poly I:C. At 6 h post-transfection, lysates were collected for analysis of the phosphorylation status of PKR by immunoblot. (B) Ectopic expression of WT or mutant σ3 in WT A549 cells does not prevent poly I:C-induced RLB formation. Cells were induced and treated as before, but at 6 h after transfection cells were fixed and immunostained for G3BP and σ3. The number of σ3 positive cells containing > 3 foci of G3BP-positive granules were counted (> 200 cells were counted per biological replicate). (C) Ectopic expression of WT or R296T mutant σ3, but not K287T prevents formation of SGs in RNase L KO cells treated with poly I:C. Data shown represent the mean ± s.d. of three independent experiments. Multiple comparison, non-paired t tests were used to analyze differences compared to mock. (ns = not significant; ***, $P < 0.001$).
(TIF)

**S13 Fig. Weights of reovirus-infected mice at times of myocarditis.** C57BL/6J 3-4-day-old mice were inoculated perorally with $10^7$ PFU of WT, K287T, or R296T virus. Mice were weighed and euthanized 8 d pi for either RNA/histology or viral titer experiments. Each symbol indicates the weight in grams of one mouse. Mean and SEM are shown. Groups were not statistically different by ANOVA (*, $P > 0.05$).
(TIF)

## Acknowledgments

We thank Faraz Ahmed and Jen Grenier for assistance with RNAseq analysis, Lauren Webb for assistance in BMDCs culture, and Andrew Brodrick for assistance with confocal microscopy.

## Author Contributions

**Conceptualization:** Terence S. Dermody, John S. L. Parker.

**Data curation:** Yingying Guo, Mercedes Lewandrowski, Shaun T. Cross, Olivia L. Welsh.

**Formal analysis:** Yingying Guo, Shaun T. Cross, Danica M. Sutherland, John S. L. Parker.

**Funding acquisition:** Terence S. Dermody, John S. L. Parker.

**Investigation:** Yingying Guo, Meleana M. Hinchman, Mercedes Lewandrowski, Shaun T. Cross, Danica M. Sutherland, Olivia L. Welsh.

**Methodology:** Yingying Guo, Meleana M. Hinchman, Danica M. Sutherland.

**Project administration:** Danica M. Sutherland.

**Software:** Shaun T. Cross.

**Supervision:** Terence S. Dermody, John S. L. Parker.

**Writing – original draft:** Yingying Guo, John S. L. Parker.

**Writing – review & editing:** Yingying Guo, Shaun T. Cross, Danica M. Sutherland, Olivia L. Welsh, Terence S. Dermody, John S. L. Parker.

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
