## [Decision Letter · Decision Letter 0]

12 May 2021

Dear Dr. Parker

Thank you very much for submitting your manuscript "The multi-functional reovirus σ3 protein is a virulence factor that suppresses stress granule formation to allow viral replication and myocardial injury" for consideration at PLOS Pathogens. As with all papers reviewed by the journal, your manuscript was reviewed by members of the editorial board and by several independent reviewers. In light of the reviews (below this email), we would like to invite the resubmission of a significantly-revised version that takes into account the reviewers' comments.

The manuscript was reviewed by three individuals with extensive expertise in reovirus biology. Overall, all three reviewers were quite laudatory in their assessment of the work, commenting favorably on the significance and quality of the work. However, the reviewers had several important questions and concerns regarding the work presented in the manuscript. These will need to be thoroughly addressed and the manuscript appropriately modified before further consideration for publication. I would expect that additional experimental data will need to be included in the manuscript to satisfy the reviewers' concerns.

We cannot make any decision about publication until we have seen the revised manuscript and your response to the reviewers' comments. Your revised manuscript is also likely to be sent to reviewers for further evaluation.

Sincerely,

John T. Patton, PhD

Associate Editor

PLOS Pathogens

Adolfo García-Sastre

Section Editor

PLOS Pathogens

Kasturi Haldar

Editor-in-Chief

PLOS Pathogens

orcid.org/0000-0001-5065-158X

Michael Malim

Editor-in-Chief

PLOS Pathogens

orcid.org/0000-0002-7699-2064

Thank you very much for submitting your manuscript "The multi-functional reovirus σ3 protein is a virulence factor that suppresses stress granule formation to allow viral replication and myocardial injury" (PPATHOGENS-D-21-00634) for consideration by PLOS Pathogens. The manuscript was reviewed by three individuals with extensive expertise in reovirus biology. Overall, all three reviewers were quite laudatory in their assessment of the work, commenting favorably on the significance and quality of the work. However, the reviewers had several important questions and concerns regarding the work presented in the manuscript. These will need to be thoroughly addressed and the manuscript appropriately modified before further consideration for publication. I would expect that additional experimental data will need to be included in the manuscript to satisfy the reviewers' concerns. be required that adequately addressing issues raised by the reviewers will require additional new data. Thank you again for submitting such an outstanding manuscript to the journal. We look forward to receiving the revised manuscript in the near future and its ultimate publication.

Reviewer's Responses to Questions

**Part I - Summary**

Reviewer #1: This manuscript explores the role of the dsRNA-binding capacity of the reovirus �3 protein in the inhibition of protein kinase R (PKR), and its subsequent effects on virus replication, cytoplasmic ribonucleoprotein (RNP) granule formation, and virulence in a newborn mouse model. The authors create and screen mutants of basic residues in the RNA-binding domain of �3, and identify mutants that lack the capacity to bind to the synthetic dsRNA analog, poly(I:C). Two such mutants in the T1L virus background, T1L-K287T and T1L-R296T, were selected for further study, as they exhibit disparate effects on single- and -multiple round replication in several cell lines, including A549 lung epithelial cells; T1L-K287T exhibits reduced replication in comparison to either wild type T1L or T1L-R296T. T1L-K287T shows increased viral mRNA levels, along with significant reductions in (-)-strand RNA and viral protein synthesis. Additionally, T1L-K287T elicits strong phosphorylation of PKR, indicative of its activation, and subsequent suppression, of cellular protein synthesis. T1L-K287T also induces significantly higher levels of TIAR+ RNP granules than either wild type T1L or T1L-R296T. Formation of these granules following infection with T1L-K287T is largely dependent upon PKR, while it appears that T1L-R296T is more prone to forming TIAR-negative granules, attributed to RNAseL (termed RNAse L bodies; RLB). These results suggest that dsRNA binding and other functions of �3 may alter the formation process for different cellular RNP granules. Importantly, the growth restriction of T1L-K287T was abrogated in PKR knockout cells, suggesting that the PKR-dependent processes induced by this virus function to limit infection. Finally, the authors examined the pathogenesis of these viruses in a peroral newborn C57BL/6J mouse model. Although all three viruses showed similar peak titers at the initial and secondary site of infection, T1L-K287T showed, generally, a slower increase in replication at secondary sites, and an earlier clearance, and its capacity to induce myocarditis was significantly reduced. RNA-seq analysis showed no significant differences in cellular gene expression caused by the three viruses in hearts of infected mice at 4 days post-infection, perhaps indicative of a more effective, but not necessarily different in kind, innate immune response.

Overall, the manuscript presents very significant, and intriguing, data, which contributes a new understanding of the role dsRNA binding versus other �3-mediated inhibitory processes in the regulation of PKR activity following reovirus infection. The data presented is well-controlled, clear, and largely supports the conclusions made by the authors. The manuscript is also very well-written and readable. A weakness of the manuscript is some assumptions made about the representativeness of poly(I:C) as a synthetic dsRNA substrate; binding of this substrate may not fully mirror the dsRNA binding capacity of the �3 protein, and it would be informative to use viral dsRNA to confirm the findings made using poly(I:C). The authors conclude, based on lack of dsRNA binding to the R296T mutant, which retains the capacity to suppress PKR phosphorylation, that �3 has separate PKR inhibitory activity, but it is possible that there are more subtle effects around the competition for dsRNA between �3 and PKR that may inhibit PKR activation, especially in the context of “native” viral RNA vs. poly(I:C). To this end, experiments aimed at assessing i) the competition for dsRNA binding between �3 and PKR; ii) the nature of poly(I:C) vs viral RNA binding; and iii) whether there are any direct interactions between �3 and PKR, would be highly informative as to the mechanism of �3 function. A second area that could be cleared up would be the role of defective particles in releasing viral RNA into infected cells. The authors speculate, in the discussion, about the role of dsRNA release from damaged incoming particles or from early transcription (line 533), in inducing a PKR response, which leads to the question of whether the mutants have a differential production of non-infectious particles, given the structural role of �3 in the reovirus virion (although the data from figure S4 suggests that they retain the same level of trypsin sensitivity, there may be other impacts on particles that could affect the outcome of these experiments).

Reviewer #2: Guo and colleagues engineered a series of reovirus sigma3 protein (σ3) mutants in the pocket reported to be involved in binding to double-stranded (ds) RNA. Using poly(IC) pulldown assays, they showed that two of these mutants, K287T and R296T, were no longer capable of binding dsRNA. Importantly, the authors ensured that the insertion of these mutations did not impair the ability of σ3 to dimerize and to form hetero-oligomers with the capsid protein µ1. They further questioned the role the K287T and R296T mutations in the context of a full virus genome. While in L929 cells both replicated to levels similar to those of the wild type virus, the recombinant viruses showed distinct phenotypes in other cell types including A549 cells. Notably, the recombinant virus K287T replicated less efficiently than the wild type virus and induced a strong activation of the stress pathways as measured by PKR and eIF2α phosphorylation levels in infected cells. This mutant also appeared to block less efficiently stress granule (SG) formation at later stages of the infection.

The manuscript is scientifically sound and experiments are well executed. Generally, the data is of very good quality and well presented. Nevertheless, some of the conclusions require additional support, including the addition of quantifications that would strengthen the message of the manuscript.

Reviewer #3: Guo and colleagues have submitted an elegant manuscript that investigates the mechanism by which reovirus sigma-3 protein promotes viral translation. The reovirus sigma-3 protein is an outer capsid protein that binds dsRNA. In this study the authors engineered a series of mutations within the putative RNA-binding domain on sigma-3. Sixteen lysine/arginine mutations to threonine were generated and tested for the ability of bind a double-stranded RNA mimic namely poly I:C. A number of mutations affected protein folding or had no effect on RNA binding. Initial studies with recombinant mutant viruses were undertaken in L929 cells. Alas the authors did not find discernible differences in one-step or multi-step infection cycles, or to cellular responses. However, when infection studies in A549, Caco-2 or BMDCs were compared, the authors observed an effect with the K287T mutant virus compared to wild-type and R296T viruses. The remaining manuscript specifically investigates the K287T and R296T mutations. Both mutations disrupted poly I:C binding, although only K287T showed a discernible phenotype. Specifically, recombinant reovirus expressing the sigma-3 K287T mutant protein resulted in decreased viral titers. Guo and coworkers systematically show the sigma-3 K287T mutation does not affect mRNA synthesis, however the abundance of viral proteins are decreased which is coincident with phosphorylation of PKR and eIF2-alpha, the formation of stress granules resulting in the decrease in replication. Notably this effect on translation and replication is negated in cells deficient in PKR. Last the authors show that this mutation within sigma-3 has a notable effect on pathogenicity, namely decreased myocarditis in neonatal mice compared to infection with wild-type or the R296T mutant virus. This is a beautiful study that builds on mutational analysis to molecular and cellular effects to disease studies. The experiments are well designed with the appropriate controls and statistics, and are systematic. As such the data are clear and the manuscript is well written.

**Part II – Major Issues: Key Experiments Required for Acceptance**

Reviewer #1: Major concerns:

1) As outlined above, testing binding to viral dsRNA may be informative as to more subtle differences in RNA binding between the mutants.

2) Although not necessary to justify the main conclusions of the manuscript, inclusion of competition assays for dsRNA binding between �3 and PKR, or testing for direct interactions between �3 and PKR, may also be highly informative to the mechanism for �3 inhibition of PKR.

3) An assessment of the particle:PFU ratio, as well other assessments of particle stability (beyond trypsin sensitivity) would help to exclude other potential confounders to the interpretation of the data.

4) Fig. 4: Given the decreased viral protein production of the T1L-K287T mutant, use of immunofluorescence for �2 is not the best marker for reovirus-infected cells (some of the cells noted as “infected” appear to have very little to no �2 staining. The authors use smFISH in subsequent figures, which could be a better proxy to label infected cells (although, given the high MOI used, restricting the analysis of granules only to infected cells may not be necessary to show an overall increase in granule-containing cells in the experiment).

5) The authors do not mention any measures of the overall effect of infection on disease status in the animals. Did any of the animals succumb to myocarditis? Did they exhibit weight loss or other pathologic signs or sequelae? A bit more characterization of the disease manifestations elicited by the mutant viruses would help in interpreting the overall pathologic impact. Along these lines, based on the figures, it appears that (perhaps with a single exception) the R296T virus may have caused even more significant myocarditis than T1L, which would be a very intriguing phenotype. More exploration of this could prove very interesting – would complete loss of dsRNA binding attenuate, but a partial effect (or retaining PKR inhibition in the absence of dsRNA sequestration) lead to a more severe disease phenotype?

Reviewer #2: 1. The fact that viral mRNA levels are increased for the mutant virus K287T although viral replication and protein synthesis are attenuated is very intriguing. The authors propose that reduced viral translation is likely due to the host translational repression and mention line 257 that this mutant induced increased p-PKR and p-eIF2α levels. However, this claim is not supported by the Western blot analysis shown in Figure 3D. While p-PKR levels are indeed clearly increased, an increase in p-eIF2α levels is not detectable. Based on this, the conclusion appears to be too preliminary. Is cellular translation repressed? To support this point, the authors should perform a ribopuromycin assay, which allows for the measurement of puromycin incorporation and thereby de novo protein synthesis, either by fluorescence microscopy or by Western blot analyses. In addition, the number of repeats and a quantification of p-PKR and p-eIF2α levels should be provided, as done for Figure 3C.

2. To determine the potential SG-suppressive function of σ3 mutant, the authors analysed the number of TIAR-positive cells in response to wild-type and mutant virus infection (Figure 4B). Some statements are unclear in this section to this reviewer. The authors mention line 271 that in the case of wild-type virus, SG formation peaks at about 6 h post-infection (hpi) and is repressed thereafter (24 hpi). In work by the Miller lab they refer to, infection induces SGs in nearly 40% of infected cells between 4 and 6 hpi. This number decreases drastically already at 12 hpi. In the histogram shown in Figure 4B, the number of SG-positive cells induced by infection with the wild-type virus remains low, less than 5%, and mostly unchanged between 6 and 18 hpi. Is this discrepancy due to the fact that SG were analysed in all cells and not only infected cells? Is it due to the use of a different cell type and therefore different kinetics? From the result shown in 4B, we can only conclude that this wild-type strain induces little or no SGs.

3. Related to the previous point. Although this reviewer agrees that more SG-positive cells are detected upon infection with the mutant virus K287T at a later time point, it cannot rule out whether this virus simply induces more SGs because of increased p-eIF2α levels. The direct evidence showing that the mutant virus suppresses SGs after their initial formation, as concluded in line 283 is missing. This result could be easily addressed by performing an experiment in which cells infected with both wild-type and mutant viruses are treated with arsenite.

4. The authors used immunofluorescence analysis combined with FISH to visualize differences in S4 mRNA abundance in viral factories (VFs) or SGs for the different viruses, for example, less S4 mRNA is found in VFs for the mutant virus K287T. While this analysis can be very informative and the images are of very good quality, it lacks quantifications to support the different claims and highlight the observed differences. We can hardly conclude on the basis of the provided representative still images. Quantification would also avoid using terms such as "often co-localized" or "mostly co-localized" that are only poorly indicative. In addition, the number of cells analysed in Figure 5 should be indicated.

5. The authors show that as described previously, TIA1 is a SG-specific marker that is not detected in RNase L Bodies (RLBs). How much does the quantification shown in Figure 6A differs when analysis TIA1-spectic SGs. This additional analysis would strengthen the difference between the K287T and the R296T mutant viruses.

6. While the mouse experiment is interesting and provides in vivo information about the mutant virus, it is not clear what this piece of data along with the transcriptomic analysis actually contributes to the SG aspect. There is no evidence of a direct link to PKR in vivo. As the authors themselves mention in line 404, one can only speculate that the reduced myocarditis is due to the initial formation of SGs by the mutant virus K287T. This could alternately be a result of the reduced level of replication. For this reason, the title should be reconsidered.

Reviewer #3: 1) The authors initially screen sigma-3 RNA binding by examining the interaction with poly I:C, a mimic of double-stranded RNA. It is notable that both K287T and R296T are deficient in binding poly I:C, yet only K287T shows a phenotype.

a. Is there a difference in the ability of the K287T and R296T mutants to interact in vitro with reovirus double-stranded RNA or in cells with viral mRNA, negative-sense RNA and double-stranded RNA during infection? Showing such a difference would provide additional support towards the proposed model.

b. Lines 118-120: the authors note that the amount of free sigma-3 (or not in complex with mu-1) might affect translational shutoff. Is there a difference in the abundance of free sigma-3 between the K287T and R296T mutants that might support this hypothesis and/or the author’s model?

2) While the authors clearly show a difference between the K287T and R296T, it would be good to include in the discussion why R296T (even though the mutant has a clear defect in binding Poly I:C) does not show the same phenotype as K287T. While this is likely speculation, or the basis of a future manuscript, this would provide additional context to the paper.

**Part III – Minor Issues: Editorial and Data Presentation Modifications**

Reviewer #1: 1) Fig. 3: Panel B is referred to first in the results text, followed by panel C, then panel A. Either relabel the figure, or reorder the results paragraph.

2) There are some tantalizing differences in RNA synthesis and protein synthesis between the mutants and wild type T1L, with only 3 replicates. Is the N high enough to really statistically power the results? Additional replicates may help to allow the authors to discuss these more subtle difference, which may, nevertheless, impact virus replication and pathogenesis.

3) Line 227/285/etc.: The authors repeatedly state that reovirus induces SG early in infected cells (citing refs. 27 and 28). These experiments were only done using T3D, and in different cell types (e.g. HeLa), and their own data suggests that WT T1L does not induce SGs at an early time in A549 cells, rather that T1L-K287T fails to continue to suppress SG formation at later times at infection. This language should be clarified, as there appear to be serotype-/cell-type specific effects that may not warrant the broad claim as a “universal” reovirus phenotype.

4) Line 295: the authors make a comparative claim, that the “level of colocalization was less than that in WT-infected cells.” There is no quantitative data to support this claim, so it should be amended, or quantification should be provided.

5) Line 302: The authors indicate that the R296T mutant still sequesters viral RNA in SGs, and that “the resultant sequestration impairs viral replication,” yet R296 does not exhibit a replication defect. Please clarify.

6) Figure 7: There appear to be some issues with the asterisks indicating statistical differences (at least in the PDF as viewed on screen, versus printing).

7) Given the (perhaps not quite statistically significant) enhancement of myocarditis by the R296T mutant, it would be interesting to have some more data on the 41 genes solely induced by this mutant (Fig. 9) – do these give any indication as to a potential phenotype? Overall, the RNA-seq data is underexplored, as there may be some useful information there; addition of a second time point (at perhaps day 8), could really shed some light on to the mechanism for the less severe phenotype induced by T1L-K287T (and potentially the more severe phenotype induced by R296T).

8) Line 475: The authors interpret the increased viral RNA levels in the T1L-K287T to the sequestration of the RNA, making it less prone to degradation/turn-over, but there are other possible interpretations. Could the dsRNA-binding capacity of �3 alter the overall transcription of the viral RNAs, perhaps as a means of preventing too much RNA synthesis, leading to a more robust innate immune response? A little more speculation here might be warranted.

Reviewer #2: 1. Figure S3: Quantification of Western blots and an indication about the number of repeats is missing.

2. Scale bars are missing in all immunofluorescence assays.

3. Figure 4C: please provide sill images of mock cells.

4. The number of cells analysed in Figures S7 and S8 is missing.

5. A growth defect of PKR knockout cells is mentioned but not really shown. Any data to support the hypothesis that this delay is rescued in double knockout cells?

Reviewer #3: The authors generated sixteen mutant proteins. Of these three mutant proteins expressed poorly, seven mutations did not affect RNA binding, and six mutations ablated RNA binding. Of those six mutants unable to bind poly I:C, why did the authors focus on K287T and R296T? The rationale for focusing on those two mutations would be good to include in the results section.

PLOS authors have the option to publish the peer review history of their article (what does this mean?). If published, this will include your full peer review and any attached files.

Reviewer #1: No

Reviewer #2: No

Reviewer #3: No
---

## [Editor Report · Decision Letter 1]

21 Jun 2021

Dear Dr. Parker,

We are pleased to inform you that your manuscript 'The multi-functional reovirus σ3 protein is a virulence factor that suppresses stress granules formation and is associated with myocardial injury' has been provisionally accepted for publication in PLOS Pathogens. The outstanding efforts made by the authors in responding to the reviewer's comments and concerns are especially appreciated.

Best regards,

John T. Patton, PhD

Associate Editor

PLOS Pathogens

Adolfo García-Sastre

Section Editor

PLOS Pathogens

Kasturi Haldar

Editor-in-Chief

PLOS Pathogens

orcid.org/0000-0001-5065-158X

Michael Malim

Editor-in-Chief

PLOS Pathogens

orcid.org/0000-0002-7699-2064
---

## [Editor Report · Acceptance letter]

1 Jul 2021

Dear Dr. Parker,

We are delighted to inform you that your manuscript, "The multi-functional reovirus σ3 protein is a virulence factor that suppresses stress granules formation and is associated with myocardial injury," has been formally accepted for publication in PLOS Pathogens.

Best regards,

Kasturi Haldar

Editor-in-Chief

PLOS Pathogens

orcid.org/0000-0001-5065-158X

Michael Malim

Editor-in-Chief

PLOS Pathogens

orcid.org/0000-0002-7699-2064